# A powder-metallurgy-based strategy toward three-dimensional graphene-like network for reinforcing copper matrix composites

Xiang Zhang [1,2], Yixin Xu [3], Miaocao Wang[3], Enzuo Liu[1,4], Naiqin Zhao[1,4,5], Chunsheng Shi [1], Dong Lin[6], Fulong Zhu [3 ✉] & Chunnian He [1,2,4,5 ✉]

Three-dimensional graphene network is a promising structure for improving both the mechanical properties and functional capabilities of reinforced polymer and ceramic matrix composites. However, direct application in a metal matrix remains difficult due to the reason that wetting is usually unfavorable in the carbon/metal system. Here we report a powder-metallurgy based strategy to construct a three-dimensional continuous graphene network architecture in a copper matrix through thermal-stress-induced welding between graphene-like nanosheets grown on the surface of copper powders. The interpenetrating structural feature of the as-obtained composites not only promotes the interfacial shear stress to a high level and thus results in significantly enhanced load transfer strengthening and crack-bridging toughening simultaneously, but also constructs additional three-dimensional hyperchannels for electrical and thermal conductivity. Our approach offers a general way for manufacturing metal matrix composites with high overall performance.

[1] School of Materials Science and Engineering and Tianjin Key Laboratory of Composite and Functional Materials, Tianjin University, Tianjin 300072, P. R. China. [2] Joint School of National University of Singapore and Tianjin University, International Campus of Tianjin University, Binhai New City, Fuzhou 350207, P. R. China. [3] School of Mechanical Science and Engineering, Huazhong University of Science and Technology, Wuhan, Hubei Province 430074, P. R. China. [4] Collaborative Innovation Center of Chemical Science and Engineering (Tianjin), Tianjin 300072, P. R. China. [5] Key Laboratory of Advanced Ceramics and Machining Technology, Ministry of Education, Tianjin University, Tianjin 300072, P. R. China. [6] Department of Industrial and Manufacturing Systems Engineering, Kansas State University, Manhattan, KS 66506, USA. ✉email: zhufulong@hust.edu.cn; cnhe08@tju.edu.cn

Driven by the worldwide needs in the key areas of energy, transportation, aerospace, and security, the demand for new structural materials with high performance has increased greatly to attain a much wider variety of properties. These materials will not only be designed into specialization for lighter weight, stronger, and tougher mechanical properties but also demonstrate additional functional roles, including conducting electricity, conducting heat, storing energy, or sensing external stimuli. In fact, researchers have never given up the efforts in searching for new materials to advance the technological development.

Owing to its unique and fascinating properties such as superior strength and elastic modulus, giant electron mobility, high thermal conductivity, excellent mechanical flexibility, and large specific surface area, graphene (Gr), a two-dimensional and conjugated honeycomb carbon structure, has been extensively exploited to integrate in ceramic, polymer or metallic matrices for achieving novel composites with outstanding mechanical, electrical, and/or thermal properties[1–3]. However, a huge challenge still lies in taking advantage of the extraordinary properties of 2D graphene in the composites because of the easy stacking and the resultant agglomeration problems of graphene nanosheets (GNSs) as well as high contact resistance between GNSs in the matrixes[4,5], thereby resulting in only moderate enhancement efficiency in the mechanical, electrical, and/or thermal properties of the graphene-based composites[6]. As work proceeds, it is becoming increasingly clear that, to fully exert its outstanding in-plane properties of graphene for macroscopic applications, the advanced architecture design of the graphene composites at multiple length scales from the atomic to the macro level is of the first priority to be required. Compared to 2D-graphene, 3D-graphene architecture in the form of aerogels/sponges[7,8] and foams[9,10] with stable backbone exhibits advantages in relieving the severe aggregation problems, so that it not only provides interlocking structure for stress transfer, but also serves as a 3D hyperchannel with extremely low inter-sheet junction contact resistance for electrons and phonons conduction[5,10]. Various types of fabrication methods have been developed to synthesize 3D graphene network reinforced polymer or ceramic matrix bulk composites with excellent mechanical properties, and/or high electrical/thermal conductivity. Generally speaking, these preparation routes could be summed up into two typical strategies: skeleton preconstruction-backfilling (strategy I) and graphene encapsulation-powder consolidation (strategy II). In skeleton preconstruction-backfilling, the freestanding 3D graphene scaffolds are first prepared by self-assembling 2D-GNSs through freeze-drying or direct growth on the 3D templates by CVD method, followed by liquid-state precursor infiltration and matrix forming process[5,7,8]. In graphene encapsulation-powder consolidation, the matrix powders are first encapsulated with GNSs on the surface and then consolidated into bulk composites, during which process the 3D graphene network takes shape by connection of the overlapped GNSs[11–14]. However, engineering of 3D graphene reinforced metal matrix composites (MMCs) still remain a great challenge. Taking Cu as an example, the strategy I was scarcely reported due to the technological difficulties of preparation. The equilibrium contact angle of Cu on graphene (graphite) was measured as about 140°[15], suggesting a non-wetting feature of Cu/graphene interface according to the Young-Dupré equation[16]. Besides, the high processing temperature which exceeds the melting point of Cu (1083 °C) could introduce massive structural defects to the 3D graphene preform[17,18]. Furthermore, the porous network feature endows graphene with a relatively small Young's modulus of less than 100 MPa, only one of ten thousandth the figure of its 2D building blocks[19]. It indicates that during the metal infiltration, the 3D graphene preform could collapse and fail to withstand the axial compressive force caused by the weight of metal. Alternatively, on the foundation of powder processing, Strategy II offers an opportunity for fabricating graphene reinforced MMCs with special structure and properties that cannot be achieved by ordinary melting-related techniques. Nevertheless, for reduced graphene oxide (RGO) nanosheets, which are the most common 2D graphene derivatives used as reinforcement, it is supposed that they have no way to directly bond together to form an integrated 3D network structure due to the limitation of the intermediate-temperature region for densification (500 °C–1000 °C). For the one thing, the temperature is far above the level that RGO nanosheets could be assembled by the Van Der Waals force and electrostatic repulsion benefited from the functional groups on their surface[20]. For another, the temperature is much lower than the level that the overlapped nanosheets fuse together through the high-temperature defect healing and carbon atomic rearrangement[21,22]. So the obstacle lies in how to effectively bond 2D-graphene into a continuous network topological architecture in order to take full advantage of the outstanding mechanical and physical properties of 3D-graphene. To date, the strategies to successfully fabricate continuous 3D-graphene/metal composites remain rare. It is imperative to develop an advanced synthesis method to solve these problems.

Herein, we develop a powder-metallurgy based strategy for fabricating 3D graphene-like nanosheet network/copper (3D-GLNN/Cu) composites for high-performance advanced structural materials, which involves the ambient-pressure rapid thermal annealing (RTA) growth of graphene-like nanosheets (GLNs) on Cu powder and the subsequent reactive hot-pressing processes. During the hot-pressing process, GLNs grown on the Cu powders are directly welded and thus construct a 3D interconnected graphene network in the composites due to the coefficient of thermal expansion (CTE)-mismatch related thermal stress between GLNs and Cu. In the constructed composites, the highly interconnected feature of 3D-GLNN could not only endow it with a much larger interfacial shear stress than 2D isolated graphene in the composites for achieving a much better load transfer strengthening capability and a remarkably higher strengthening efficiency, but also greatly reduce the electrons scattering in the interfacial areas and construct extensive conducting highways throughout the matrix for electrons transportation. As a result, the 3D-GLNN/Cu composite demonstrates superior mechanical properties, electrical and thermal conductivity simultaneously, which has the potential to satisfy many special applications such as lightweight macroscopic conductors and heat-sinks in electronics. Moreover, this feasible and scalable bottom-up concept to welding graphene into a continuous network architecture during powder consolidation can enable new paths to design 3D network structure constructed by 2D building blocks in the metal matrix composites without the ubiquitous restrictions of currently-used melting-related processing methods.

## Results

**Fabrication of 3D graphene-like network in copper.** The first task to achieve a continuous 3D-GLNN in the Cu matrix composites is to solve the problem of uniform coating of GLNs on the surface of Cu powders which act as building blocks for constructing network architecture. In our strategy, we proposed an in-situ growth route, which starts from controllable synthesis of GLNs on Cu powders by using an ambient-pressure RTA method. Thereby, the overall fabrication process of 3D-GLNN/Cu composites could be primarily divided into three steps (Fig. 1). For the first step, a uniform coating of GLNs was in-situ grown on the Cu powder surface by a typical RTA process at 800 °C, at which temperature the sintering between Cu powders just started and a loosely-packed-sphere structure could maintain well (Fig. 2a).

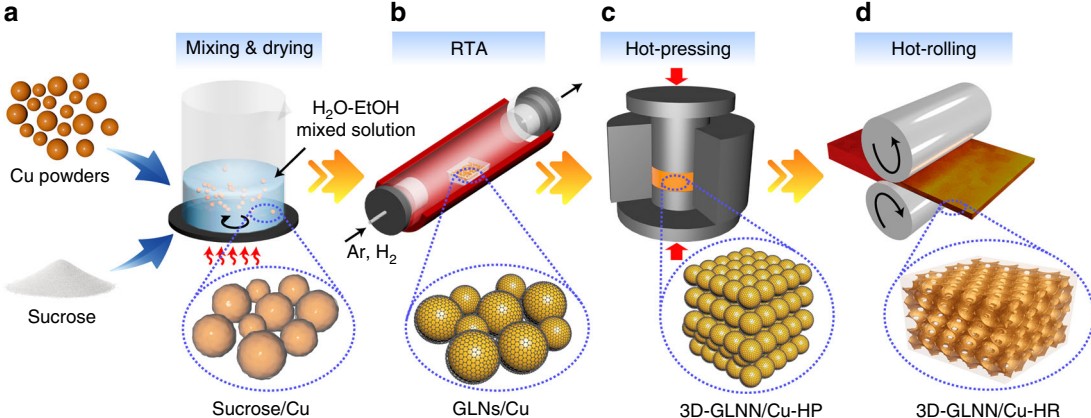

**Fig. 1 Schematic illustration of overall fabrication processes. a** The Cu powders were first coated with sucrose as a hybrid precursor. **b** The hybrid precursor was then subjected to RTA process for growing GLNs. **c** The GLNs were interconnected into a continuous network structure in the Cu matrix by using hot-pressing. **d** The fully-densified 3D-GLNN/Cu bulk composites were fabricated by hot-rolling.

Otherwise, if the RTA temperature rose to 900 °C, the Cu powders were severely sintered together and lost the powder character, which would be detrimental to the formation of GLNs (Supplementary Fig. 1). Thanks to the ultrathin and intact layer of the impregnated sucrose as the carbon source, the RTA-resulting GLNs were identified with a uniform coating structure around Cu powders (Fig. 2b). The template-dependent morphology of GLNs could be clearly spotted after etching Cu powders by FeCl$_3$/HCl solution (Fig. 2c). From the magnified image of the edge area in Fig. 2d and Supplementary Fig. 2, the GLNs demonstrate a few-layered graphene-like feature, resulting in an ultralow loading content of GLNs of 0.096 wt. % (or 0.387 vol. %) in the GLNs/Cu composite powders on the basis of C-S analysis. In the second step, the continuous network of 3D-GLNN was in-situ constructed during the hot-pressing process and the Cu matrix composite was sintered and densified simultaneously. As shown in Fig. 2e, it is apparent that an intact network structure of 3D-GLNN exposed after etching the superficial Cu matrix. The average pore size (0.5–2 μm) of the carbon skeleton matches well with the particle size of Cu powder template. Furthermore, Transmission electron microscope (TEM) image of Fig. 2f verifies that the carbon skeleton is a porous network structure constructed by the interconnected GLNs on the basal plane and forms Y-type junction. The distinctive welding feature could be spotted in Fig. 2g from the cross-section view. Two adjacent GLNs with different lattice orientation, namely A and B, converged into one singe multi-layer GLNs (A + B) with uniform lattice orientation. Taking advantage of the contrast difference between GLNs and copper in the second electron mode of SEM, a focused ion beam (FIB) 3D reconstruction technique was performed and testified the indeed formation of continuous 3D network architecture of GLNs in the Cu matrix composites (Fig. 2h and Supplementary Movie 1 and 2). In the final step, the hot-pressed 3D-GLNN/Cu bulk was treated with a multi-step hot-rolling process for further densification. The continuous network of 3D-GLNN remained intact even after a severe rolling deformation of 70% reduction in thickness, verified by SEM images of the etched surface morphology (Fig. 2i and j) and 3D reconstruction results (Fig. 2k and Supplementary Movie 3 and 4). It is known that for the commonly-used 2D graphene derivatives RGO nanosheets, residual oxygen-containing functional groups were inevitably preserved on the surface which make them undergo irreversible aggregation problems in mixing with Cu powders. As a result, even under a similar sequence of preparation steps, it led to a non-uniform distribution of isolated RGO nanosheets with totally different microstructure (Supplementary Note 1). The dramatic

difference attracted us to proceed to explore the formation mechanism of our continuous 3D-GLNN in the metal matrix.

**Formation mechanism of 3D network structure.** A fundamental understanding of the formation mechanism of continuous 3D-GLNN in Cu is of vital importance to this ground-breaking synthesis strategy. Results from initial experiments indicate that the formation of 3D-GLNN should be related to a special sheet welding mechanisms. To this end, we have carried out both experimental verification and modeling to identify the GLNs welding phenomena as well as construct 3D-GLNN/Cu with optimized network architecture.

During the hot-pressing process, parameters such as pressure, holding temperature would have great influence on the formation of the continuous network of 3D-GLNN. On one hand, the sufficient pressure and holding temperature could ensure the full sintering of Cu powders in the Cu–Cu contact area. On the other hand, the densification achieved by the high pressure and high temperature could facilitate an intimate contact between the 3D-GLNN and the adjacent Cu. In order to disclose this speculation, a series of contrast experiments were conducted in which various samples with different synthesis parameters designated as "RTA temperature-holding temperature-pressure" were prepared and characterized. As shown in Fig. 3a, the results indicated that the 800-800-50 sample possessed the most regular and homogenous 3D-GLNN structure. The magnified SEM image gave a strong evidence that the two GLNs were converged into a whole in the overlapping area. The continuous feature of 3D-GLNN was also confirmed from the Raman mapping image in Fig. 3g. Without exerting pressure, the loosely-packed GLNs/Cu sample (800-800-0) could not be sintered efficiently and only resulted in a collapsed and layer-stacking structure of GLNs exposed on the etched surface (Fig. 3b). The holding temperature is another sensitive parameter that dominates the sintering process by its influence on the diffusivity and plastic flow of Cu matrix. The low holding temperature could not offer enough driving force for Cu powder to overcome the obstacles for sintering. Therefore, the 800-400-50 sample which was hot-pressed under 400 °C had a poor densification and it only led to an irregular 3D-GLNN structure in copper (Fig. 3c). Further to say, the magnified SEM image suggests that only a small part of the overlapping area were welded together between layer A and B, thus it merely formed an X-type contact in the junction area. From the point view of chemical structure, XPS is suitable for identifying the $sp^3$ to $sp^2$ ratio and chemical bonds changes in the 3D-GLNN exposed on

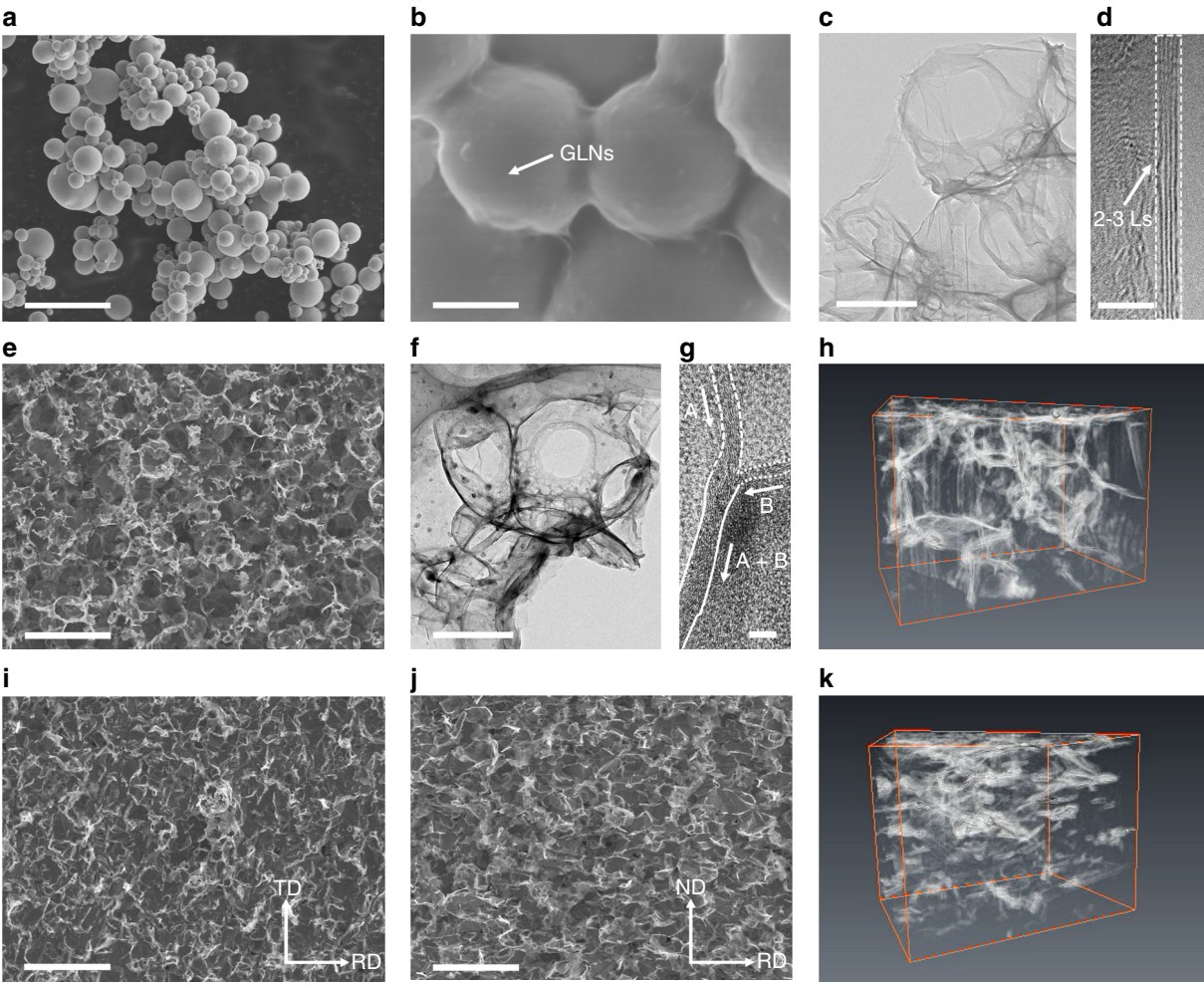

**Fig. 2 Characterizations of composite powders and bulk materials. a**, **b** SEM image of the as-grown GLNs/Cu composite powders with a loosely-packed-sphere structure (**a**) and the typical morphology of uniform GLNs coating structure around Cu powders (**b**) synthesized at 800 °C. Scale bar, 5 µm (**a**); 200 nm (**b**). **c**, **d** TEM image of GLNs after etching Cu powders (**c**) and the corresponding high-resolution TEM (HRTEM) image of the edge area (**d**), in which the thickness of GLNs was determined as 2–3 layers. Scale bar, 200 nm (**c**); 2 nm (**d**). **e**, **f** SEM image (**e**) and TEM (**f**) image of 3D-GLNN after Cu etching in the hot-pressed 3D-GLNN/Cu composites. Scale bar, 5 µm (**e**); 200 nm (**f**). **g** HRTEM image of the Y-type interconnection area of 3D-GLNN, where the layer A and layer B merged into layer A + B. Scale bar, 5 nm. **h**, **k** Snapshot of FIB-3D reconstruction results of 3D-GLNN in the **h** hot-pressed 3D-GLNN/Cu (model size:3.85 × 2.14 × 2.00 µm) and **k** hot-rolled 3D-GLNN/Cu (model size:3.51 × 2.10 × 2.08 µm). **i**, **j** SEM images of 3D-GLNN after Cu etching in the hot-rolled 3D-GLNN/Cu from **i** TD-RD plane and **j** ND-RD plane. Scale bar, 5 µm (**i**, **j**).

the surface of bulk materials. Generally, the deconvoluted C1s peaks of GLNs could be fitted into four peaks: $sp^2$ (284.6 eV), $sp^3$ (285.1 eV), C–O (286.1 eV) and C=O (288.3 eV)[23]. The ratio of the composing bond types were calculated from the integrated area of the fitted peaks (Supplementary Table 1). The significantly increased $p_{sp^3}/p_{sp^2}$ ratios of 800-800-0 (0.83) and 800-400-50 (14.3) compared to that of the RTA-800 composite powders (0.19) indicated that the insufficient hot-pressing conditions caused structure damage to GLNs. To figure out whether the crystallinity of the in-situ grown carbon materials contributes to the formation of 3D-GLNN structure, we moved on to prepare bulk composites with low-temperature (400 °C) RTA-obtained powders for comparison. The lower annealing temperature in RTA process led to an incomplete decomposition of sucrose and thus a high ratio of amorphous carbon and residual oxygen functional groups on the surface (Supplementary Fig. 4). From Fig. 3d, it is interesting to note that the 400-800-50 sample exhibited a network structure analogy to that of 800-800-50 sample. However, difference could be spotted in the magnified image of the selected areas that the X-type contact junction

existed between two adjacent layers in 400-800-50 samples. It is no doubt that this phenomenon suggested that the welding was localized in a small part of the composites. By comparing the XPS result in Fig. 3e and Supplementary Fig. 4, the $p_{sp^3}/p_{sp^2}$ ratio of GLNs in 400-RTA significantly decreased after hot-pressing. Meanwhile, the Raman results in Fig. 3f indicated that the structural disorder of carbon species, which could be determined by the ratio ($I_D/I_G$) between D band (1350 cm$^{-1}$) and G band (1580 cm$^{-1}$)[24], also decreased after hot-pressing for 400-RTA composite powders. This could be strong evidence that structure healing occurs during the formation process of 3D-GLNN. And not coincidentally, the Raman analysis result also demonstrated a similar structural healing during hot-pressing of RGO/Cu composite as the $I_D/I_G$ ratio of the bulk decreased compared that of RGO/Cu powders (Supplementary Fig. 5). According to the analysis above, it is reasonable to suggest that the insufficient driving force during hot-pressing in 800-800-0 and 800-400-50 samples could not overcome the obstacles for Cu densification, which hindered the structure healing process and failed to construct an intimated bonding between adjacent GLNs. Instead,

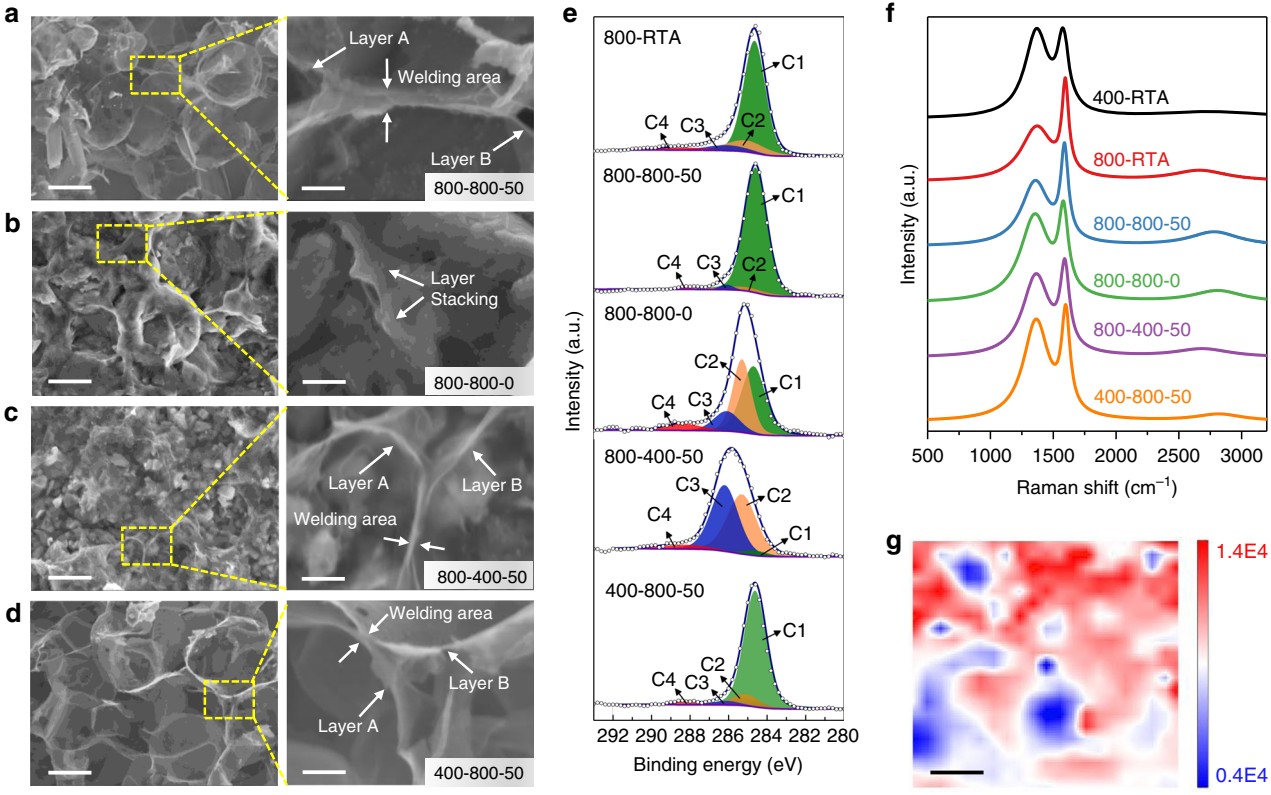

**Fig. 3 Sructure evolution during hot-pressing. a–d** SEM images of the reinforcement morphology after etching the surface Cu layer in **a** 800-800-50, **b** 800-800-0, **c** 800-400-50 and **d** 400-800-50. Scale bar, 1 μm for the left panel and 200 nm for the right panel (**a–d**). **e, f** Deconvoluted XPS C1s spectrum and (**f**) Raman spectra of 3D-GLNN in the composite powders (400-RTA and 800-RTA) and bulks (800-800-50, 800-800-0, 800-400-50 and 400-800-50). **g** Surface Raman map of 800-800-50, the peak density of the 3D-GLNN G band (1580 cm⁻¹) is imaged in red. Scale bar, 2 μm (**g**).

the $sp^2$ structure of GLNs could be subjected to unexpected damage due to the mechanical force or gas impurities absorbed in the gaps of Cu powders. In addition, by comparing the results using Cu powders in different shapes and sizes (Supplementary Figs. 6 and 7, and Supplementary Note 2), the successful construction of 3D-GLNN/Cu requires the Cu powders to be in regular spherical shape and have relatively small size to facilitate partial sintering between Cu powders during GLNs synthesis and thus form the interpenetrating graphene-like network structure afterward.

The matrix effect could be another important factor for the formation of an interconnected graphene network architecture. The high CTEs of metallic materials endow them with high thermal-elastic properties, which dramatically influence the synthesis and phase transformation process[25]. It is well recognized that the huge gap of CTEs between the matrix and the reinforcement could cause high internal stress in the composites during the rapid cooling after hot processing. As previously confirmed, it may result in massive geometric necessary dislocations (GNDs) accumulated on the interface and thus strengthen the materials[26]. However, rare studies have been focused on the interaction between reinforcement and the matrix considering the thermal stress effect during the high-temperature processing. Based on the thermal-elastic mechanism, the peak thermal stress ($\sigma_{th}$) of the 2D film-substrate model could be calculated as,[27]

$$\sigma_{th} = \frac{E_c}{1-\nu}(\alpha_s - \alpha_c)(T - T_0) \qquad (1)$$

where $E_c$ and $\nu$ are the elastic modulus and the Poisson ratio of the film, respectively. $\alpha_s$ and $\alpha_c$ each represents the CTE of the substrate and film. $T - T_0$ is the temperature gradient. Taking Cu–Gr/Gr–Cu

model as an example, the instantaneous peak thermal stress on the interface could exceeds 10 TPa for an 800 °C temperature gradient. It is no doubt that the enormous thermal stress which is dozens of times higher than the external press applied could cause a big impact on the embedded GNSs. In order to gain insights into the effect of Cu matrix on the welding process of GLN layers, it is particularly insightful to examine the evolution of the micro-structure of GLNs embedded in the copper matrix. The molecular dynamics (MD) simulations were conducted based on a simplified structural model consisted of two 3-layer graphene/copper matrix (3LGs/Cu) countering parts with face-to-face contact, as depicted in Fig. 4a. The initial space between the two top graphene layers was set as 3.35 Å to mimic the contact feature in the hot-pressing process under sufficient pressure and temperature. To approach the real structure of GLNs in the experiments, the graphene layers were designed into an imperfect structure with a defect concentration of 0.5 at. % by artificial creating single-atom vacancies on the surface (Supplementary Fig. 9a). Besides, to facilitate the observation of the interaction between Cu and graphene on the interface, we fixed the borders of the model in the Z and Y direction and imposed the periodic boundary condition in the X direction. The structure of the model relaxed to thermal equilibrium under NVT ensemble at 1293 K was verified with good stability. The simulation results under different loading temperature at 493 K, 693 K, 893 K, and 1093 K were demonstrated in Fig. 4b. A downward trend of the distance between graphene layers could be observed with the increase of the loading temperature, which is caused by the expansion-induced thermal stress of Cu matrix. To visualize the bonding conditions at different temperatures, a lower-limit length of the formation of covalent-like bonds were set as 2.80 Å. It is well-known that the interlayer binding energy increased as the distance

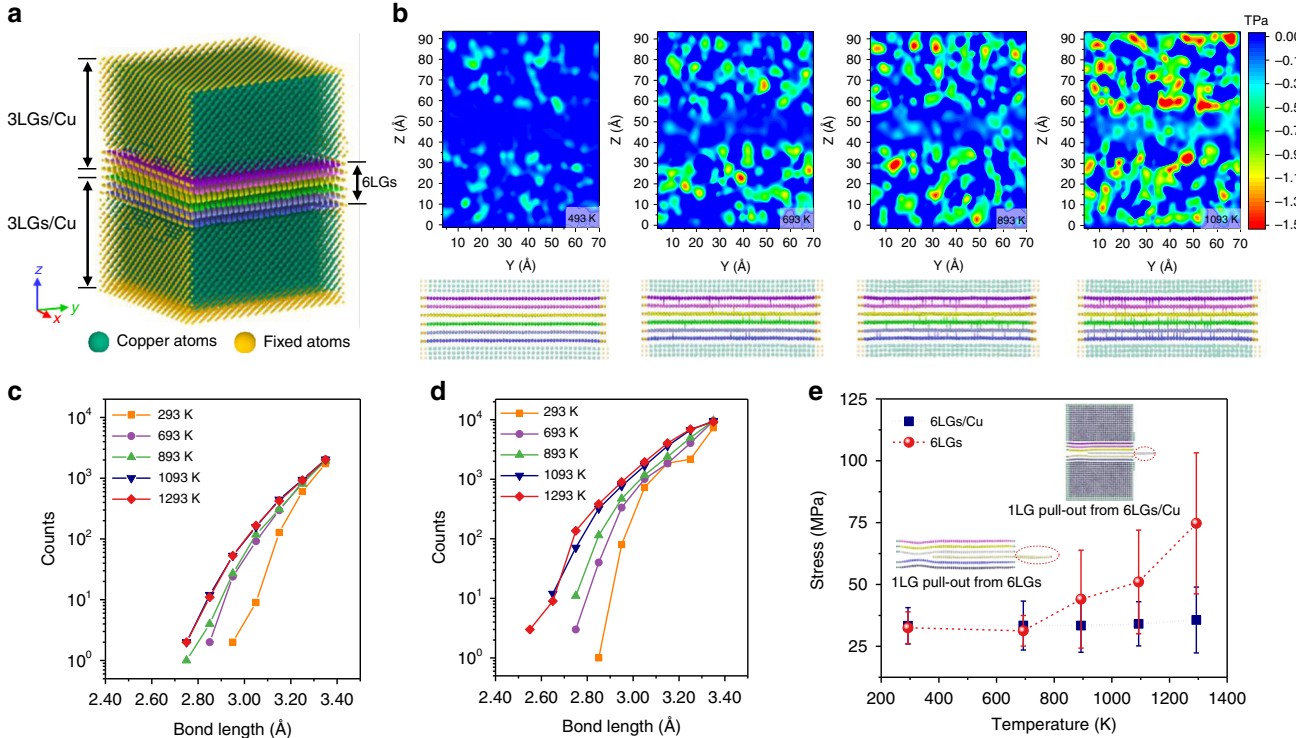

**Fig. 4 MD simulations of welding between graphene-like nanosheets. a** Atomic configuration of two face-to-face contact 3LGs/Cu (6LGs/Cu). **b** Simulation results illustrating the color Z-direction stress maps of Y–Z single-layer slice in the middle of 6LGs/Cu and the formation of covalent-like bonds between graphene layers at 493 K, 693 K, 897 K, and 1093 K. **c, d** The bonding length statistics at different simulated temperatures of (**c**) 6LGs and (**d**) 6LGs/Cu models. **e** The 1LG pull-out stress versus temperature plots for 6LGs and 6LGs/Cu. Source data are provided as a Source data file.

between graphene layers decreased to a certain level[28]. As revealed from the result, intuitively the number of carbon atoms connected by the covalent-like π−π bonding between adjacent layers increased with the rising temperature in 6LGs/Cu, suggesting an enhanced welding feature between graphene layers (Fig. 4b). Quantitatively, the number of inter-layer bonds with bond length of 2.5–3.4 Å were counted for both 6LGs/Cu and 6LGs without Cu matrix. A significant increase in the total numbers of inter-layer bonds below 3.4 Å as well as the number of short-distance bonds could be found in 6LGs/Cu compared with those of 6LGs, respectively (Fig. 4c and d). What's more, the detailed microstructure evolution proved that during the simulation a structural rearrangement occurred for the atoms near the defects in the graphene layers (Supplementary Fig. 10). In the previous study reported by Liu et al.[29], the welding between graphene layers required a high driving force provided by the high loading temperature of more than 1200 °C. While in this study, the thermal stress caused by the large gaps of CTEs between Cu and graphene promoted the compaction of graphene layers and resulted in the covalent-like π−π bonding of interlayer carbon atoms at a relatively lower temperature. The thermal stress on the graphene-Cu interface increased linearly with the rise of temperature, and thus it resulted in the rising numbers of covalently-bonded carbon atoms (Supplementary Fig. 8). Without Cu, few carbon atoms with covalent-like bonds could be observed even under a high loading temperature of 1293 K. It should be pointed out that, negligible difference was observed between the inter-layer bonding conditions of different defect concentration levels (0.5 at.%, 1.0 at.%, and 2.0 at.%) in 6LGs/Cu model (Supplementary Fig. 9b), which implies that the bonding trend is not drive by defects at a relatively low defect level. Next, we performed graphene pull-out simulations to compare the adhesion strength between graphene layers of the aforementioned models relaxed under different loading temperatures. The average pull-out stress-displacement at different

temperatures were plotted in Fig. 4e. It is demonstrated that the pull-out strength increased with the loading temperature rising above 693 K, demonstrating an enhanced adhesion strength between graphene layers. The pull-out stress between models with or without copper were further compared. It proved that under the low temperature of 293 K and 693 K, the ultimate strength of the models were almost equal due to the inappreciable contribution of thermal stress to the adhesion between adjacent graphene layers. While the pull-out strength exhibited much higher values for the models with Cu than those without Cu at a higher temperature of 1093 K and 1293 K. Therefore, the above results validate that the thermal stress during hot-pressing undoubtedly has a significant effect on the welding between the graphene layers.

Based on the analysis above, the formation of 3D-GLNN is made possible by synergy between appropriate hot-pressing parameters and the Cu matrix effect. A sufficiently high level of hot-pressing pressure and holding temperature guarantee that the in-situ grown GLNs are embedded in the grain boundaries of densified Cu matrix and reach an effective distance for interaction. Thus, 3D-GLNN forms by the thermal-stress induced sheet welding between GLNs at high temperatures.

**Composite microstructure characterization.** The continuous 3D graphene network architecture has a significant impact on the multi-scale microstructure of the copper matrix. Macroscopically, the average copper grain size was limited to a small value of 1.9 μm (Supplementary Fig. 11d) due to the grain constraint of 3D-GLNN after hot-pressing, while conspicuous grain growth effect (average grain size of 18.2 μm) was identified in the pure Cu matrix (Supplementary Fig. 11a–c). The grain structure maintained well with no obvious texture after multi-step hot-rolling as confirmed by the Electron back-scattered diffraction (EBSD)

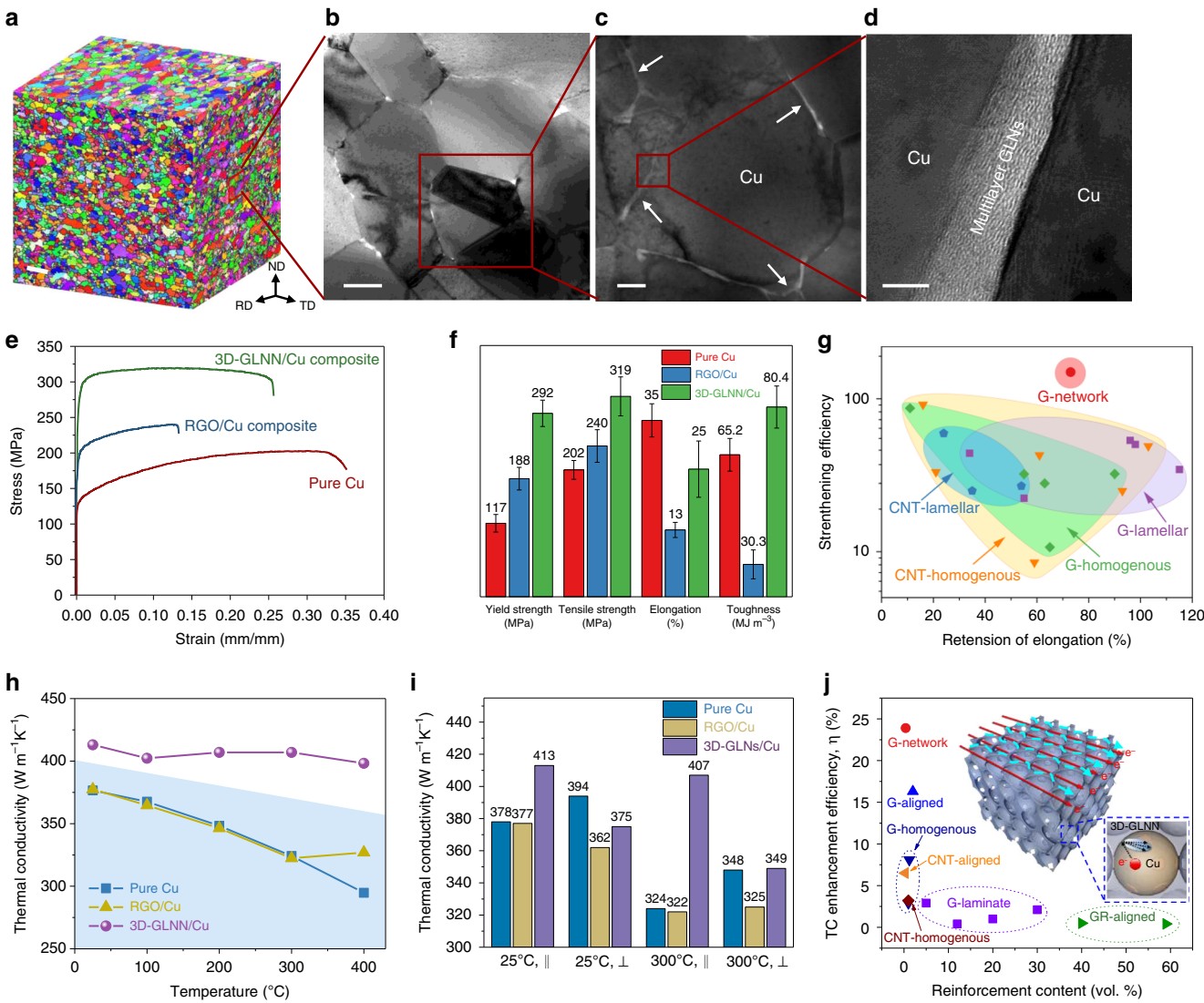

**Fig. 5 Typical microstructures and performance. a** EBSD micrographs of hot-rolled 3D-GLNN/Cu from three orthogonal directions. Scale bar, 5 μm. **b** Bright-field TEM image of the 3D-GLNN/Cu composites. Scale bar, 500 nm. **c, d** Bright-field scanning transmission electron microscope (STEM) image of the 3D-GLNN/Cu composites (**c**) and the corresponding HRTEM image taken near the grain boundaries of the composite (**d**). Scale bar, 200 nm (**c**); 5 nm (**d**). **e** Tensile stress-strain curves of pure Cu, RGO/Cu, and 3D-GLNN/Cu. **f** Comparative bar chart of mechanical properties of pure Cu, RGO/Cu, and 3D-GLNN/Cu. **g** Retention of fractural elongation versus strengthening efficiency of tensile strength plot, showing that the as-prepared 3D-GLNN/Cu composite had an outstanding combination of strengthening efficiency and ductility. **h** The in-plane thermal conductivities of pure Cu, RGO/Cu, and 3D-GLNN/Cu. **i** Comparative bar chart of thermal conductivities of pure Cu, RGO/Cu, and 3D-GLNN/Cu at room temperature (25 °C) and 300 °C. **j** TC enhancement efficiency versus graphene content plot, demonstrating a high enhancement efficiency of 3D-GLNN in thermal conductivity at a relatively small graphene content compared with other reported nanocarbon reinforced Cu matrix composites. Inset is the illustration of the electrons and phonons transfer channels in 3D-GLNN and the reduction of electrons scattering in the Cu matrix near the interface.

analysis from three inter-perpendicular orientations (Fig. 5a and Supplementary Fig. 11e, f). In combination with the morphology characterization of 3D-GLNN in the Cu bulks in Fig. 2e, h–k, the result confirms that 3D-GLNN is a flexible network, which could deform simultaneously with the matrix and keep bound to the matrix grains. In contrast, the RGO nanosheets, failed to hinder the matrix grains effectively during hot-rolling due to the limitation of its 2D shape and relatively small size (Supplementary Fig. 12a). The Talyor factor maps of the EBSD results were also provided in Supplementary Fig. 12b and c. Taylor factor is an important factor for evaluating the deformability of polycrystalline grains, which is a function of the grain orientation and the external deformation types. The larger Taylor factor values, the more plastic work needed for plastic deformation due to the increased obstacles for the movement of more slip systems[30]. A

majority of Cu grains with high Taylor factors could be spotted in hot-rolled RGO/Cu from Supplementary Fig. 12b, indicating that plastic deformation is hard to be started in the material. While in hot-rolled 3D-GLNN/Cu the grains with high Taylor factors dispersed more homogenously in the matrix, suggesting a much smaller resistance for plastic deformation. On the microscopic level, The Y-type junction feature shown in Fig. 5b, c and Supplementary Fig. 13 is consistent with the TEM image in Fig. 1g, proving that the 3D network-shaped of 3D-GLNN distribute along the grain boundaries. From the HRTEM image of Fig. 5d, the multi-layer structure obtained from the welding of the adjacent GLNs layers could be identified. In comparison, RGO were characterized with an isolated island-like distribution feature in the hot-rolled samples (Supplementary Fig. 14). Dislocation forests could be observed near the graphene/Cu interfaces in both

samples, which gave an evidence of dislocation pinning effect of graphene[31].

**Mechanical performance**. Figure 5e demonstrates the tensile response of the pure Cu matrix, 3D-GLNN/Cu, and RGO/Cu composites, where the tensile direction was parallel to the rolling direction (the detailed process of the tensile deformation process was discussed in Supplementary Note 3 and Supplementary Fig. 15). Encouragingly, 3D-GLNN/Cu delivers a yield strength (YS) of 292 ± 21 MPa, an ultimate tensile strength (UTS) of 319 ± 30 MPa, which was markedly superior to pure Cu matrix (YS: 117 ± 14 MPa, UTS: 202 ± 15 MPa). Besides, the fractural elongation (FE) of 3D-GLNN/Cu was measured as 25.4 ± 5.6%, which is only 27% reduction compared with pure Cu. Therefore, the combination of a high UTS and a moderate FE of 3D-GLNN/Cu resulted in a 23% increase in toughness (measured by integration of the area under the stress/strain curve) than pure Cu. What's more, the tensile response of 3D-GLNN/Cu also exhibits a substantial improvement in the strength (YS and UTS) and TE over RGO/Cu. The toughness calculated of 3D-GLNN/Cu is more than 1.5 times higher than that of RGO/Cu (Fig. 5f). Up to now, a large number of nanocarbon (CNTs and graphene) reinforced MMCs have verified the strengthening potential of these nano-sized reinforcements. To directly evaluate and compare the mechanical performance of these materials, we summarized the data reported by various research groups on the strengthening efficiency of UTS $(R_\sigma = \frac{\sigma_{UTS}(C)-\sigma_{UTS}(M)}{f\sigma_{UTS}(M)})$ and retention of FE (Supplementary Table 2) as plotted the scatter diagram in Fig. 5g. Thanks to the unique continuous network architecture, 3D-GLNN/Cu obtained a high strengthening efficiency of 149.7 with a low volume fraction of 0.387 vol. % 3D-GLNN. What could be observed from the figure is that for most cases of the trade-off correlation between the two parameters still exist in composites with homogenous and lamellar architecture. Remarkably, the 3D-GLNN/Cu in this study outperformed most other nanocarbon reinforced MMCs in terms of the combination of strengthening efficiency and ductility, far outside the domains enclosing the data for different architecture types reported in the literature (red star mark in Fig. 5g).

**Thermal and electrical conductivity**. Figure 5h shows the thermal conductivities (TCs) of pure Cu, RGO/Cu, and 3D-GLNN/Cu composites parallel to the rolling direction (in-plane TCs, $K_\parallel$). The TC of 3D-GLNN/Cu reached 413 W m$^{-1}$ K$^{-1}$ at room temperature which exceeded that of pure Cu (378 W m$^{-1}$ K$^{-1}$) and decreased slightly to 407 W m$^{-1}$ K$^{-1}$ at 300 °C. While a significant decline trend could be observed in the TC values of pure Cu as temperature rose from room temperature to 400 °C. A similar decreasing trend could be observed form the curve of RGO/Cu composite. The discontinuous RGO distribution resulted in lower TC values compared with those of pure Cu in the temperature range of 25–300 °C. The high-temperature thermal conductivity is more valuable for reference in considering the actual hot-wall contact condition for using copper related materials as heat transfer. Experimentally, the TC values (K) were determined by parameters of their thermal diffusivities (TD, $\alpha$), specific heat ($C_p$) and density ($\rho$) values, as shown in Eq. (2)[32]:

$$K = \alpha C_p \rho \qquad (2)$$

Accordingly, TD is the determining factor for the TC value of a particular bulk material as the value of density and specific heat were almost unchanged with the increase of temperature (Supplementary Table 3). The in-plane TDs versus temperature data were plotted in Supplementary Fig. 16. It confirms that 3D-

GLNN/Cu composite has higher room-temperature in-plane TD value of 112.706 mm$^2$ s$^{-1}$ than that of RGO/Cu (109.447 mm$^2$ s$^{-1}$) and pure Cu (107.082 mm$^2$ s$^{-1}$). The higher TD value indicates that the 3D-GLNN/Cu is more capable in conducting thermal energy than RGO/Cu and pure Cu. Excluded the negative grain-size effect[32], the macroscale enhancement of the thermal conductivity in a 3D-GLNN/Cu bulk composite can be mainly attributed to the network architecture and continuity of graphene in the copper matrix. Notably, 3D-GLNN/Cu bulk composite has an interpenetrated network, suggesting the electron heat transfer of copper remains intact. In addition, the network architecture composed by graphene stitched to each other covalently provides much more channels for phonons to transfer even at a small graphene content of 0.387 vol. %. While for the RGO/Cu, the irreversible aggregation problem could reduce the effective volume fraction of RGO nanosheets and the high heat resistance caused by sheet-overlapping degraded the TC value enhancement efficiency dramatically. The conclusion above could also be verified from the measured TC values of the aforementioned samples in the direction perpendicular to the rolling plane (through-plane TCs, $K_\perp$) (Fig. 5i). It demonstrates that 3D-GLNN/Cu has a lower $K_\perp$ value of 375 W m$^{-1}$ K$^{-1}$ than pure Cu (394 W m$^{-1}$ K$^{-1}$) at the room temperature. However, the value of TC at 300 °C for the composite decreased to 349 W m$^{-1}$ K$^{-1}$, equaling the value of pure Cu (350 W m$^{-1}$ K$^{-1}$). Generally, the rising temperature increases the probability of electron scattering at the grain boundaries in the matrix[33]. In this study, the thermal conductivity decreased mainly due to the complex interplay of both the intrinsic and boundary scattering mechanisms of copper. Thanks to the distribution of 3D-GLNN in the grain boundaries, a reduced grain boundary scattering for electrons as well as surface scattering of electrons and phonons could be achieved in 3D-GLNN/Cu at elevated temperatures[33,34]. What's more, by comparing the measurement results of 3D-GLNN/Cu and RGO/Cu, significant increase in both of the in-plane and through-plane TC values could be obtained in 3D-GLNN/Cu at different elevated temperatures, which serves as strong evidence of the successful construction of a heat conducting network by 3D-GLNN in the Cu matrix. Meanwhile, the flexible feature of the network composed by nanoscale reinforcements resulted in a slight reduction in CTE of the bulk composites, compared with pure Cu and RGO/Cu (Supplementary Fig. 17). Similarly, we defined the TC enhancement efficiency ($\eta = \frac{K_C - K_M}{fK_M \cdot 100} \times 100\%$) to evaluate the enhancement in the in-plane TC per volume fraction. By putting together the $\eta$ versus $f$ data of copper matrix composites reinforced by naonocarbon materials (typically CNTs and graphene) from the reported papers and this work, it is clear to see that the 3D-GLNN/Cu with network architecture has the highest $\eta$ at a relatively small reinforcement content (Fig. 5j and Supplementary Table 4). At the same time, the ratio between $K_\parallel$ and $K_\perp$ ($K_\parallel/K_\perp$) of 3D-GLNN/Cu (0.91) at room temperature is much higher than the values of copper matrix composites with alignment or laminate architecture. It suggests that network architecture also benefits to modify the anisotropy in thermal conductivity as well. Additionally, the electrical conductivity (EC) was measured as 60.0 ± 0.5 MS m$^{-1}$, much higher than that of pure Cu (58.5 ± 0.5 MS m$^{-1}$) and RGO/Cu (55 ± 1.0 MS m$^{-1}$). It is unanticipated to discover that this EC value even exceeds 3.4% of the International Annealed Copper Standard (IACS) (58.0 MS m$^{-1}$). The high EC of 3D-GLNN/Cu provided tangible evidence of the successful construction of the continuous electron pathway provided by the coupling between 3D-GLNN and Cu[35]. Moreover, based on the electron-transfer-related Widemann-Franz law $K/\sigma = LT$[32,36], where $\sigma$ is the EC and $L = (\pi^2/3)(k_B/q)^2 \approx 2.44 \times 10^{-8}$ W$\Omega$K$^{-2}$ is the Lorenz number, the predicted TC could explain 94% of the total measured value. It

confirms that the interconnected Cu matrix still acts as the dominant role in deciding the TC and EC of the composites at room-temperature.

## Discussion

In the previous research on polymer/ceramic matrix composites reinforced by 3D graphene, the achievement of a high strength and a high toughness have been commonly reported[5,10,37]. However, this phenomenon is not well understood and has been attributed to the network architecture of which toughening is interpreted by the crack blocking effect, namely crack bridging and crack deflection. While in this kind of materials, strengthening is generally interpreted by the simplified load transfer effect between 2D graphene and the polymer/ceramic matrix. The interfacial deformation behaviors, which microscopically dominate the interaction between 3D graphene and the matrix, were rarely studied. In particular, for metallic materials which are capable of better deformation ductility than polymers and ceramics, the micro-scale interfacial behaviors could be the key roles in the overall mechanical performance. Experimentally, the interfacial shear stress derived from the simplified shear-lag model for 3D-GLNN/Cu was estimated as 86.0 MPa, while that of 2D-RGO/Cu was calculated as 26.4 MPa (Supplementary Note 4). The result suggests that different interfacial deformation mechanisms should exist in the two systems. To deeply understand the interfacial deformation behaviors of 3D graphene in the composites, in situ SEM tensile test were performed in comparisons with 2D graphene/Cu. During the test, dynamic observation of the whole cracking process could be performed on the dog-bone samples preprocessed with a notch (Supplementary Fig. 18). Figure 6a presents the load-displacement curves of 3D-GLNN/Cu and RGO/Cu with several interrupted stages. The SEM capturing of the crack evolution of the stages (I–VII) of 3D-GLNN/Cu and (A-D) of RGO/Cu in Fig. 6a was shown in Supplementary Fig. 19 and 22, respectively. During the early stage of loading, the microcracks initiated from the pre-made notch propagated slowly until reaching the stage near the yield strength point (III) for 3D-GLNN/Cu (Supplementary Fig. 20) and (B) for RGO/Cu (Supplementary Fig. 22b). After that, the crack propagated steadily until the final fracture of the samples. It is found that the 3D-GLNN/Cu has a higher tensile strength as well as a longer elongation compared with those of RGO/Cu, suggesting a good fracture tolerance capability. The capturing of the two composites after the final fracture are shown in Fig. 6b, c. The result reveals that only one principle crack extended through the sample with no obvious secondary cracks, indicating that ductile fracture happened in both composites[38]. During the propagation process, it is hard to spot RGO nanosheets on the side position of cracks due to their random distribution in the matrix. Besides, it turned out that RGO nanosheets were peeled off from the matrix based on the SEM images in Fig. 6d and Supplementary Fig. 23. It tells a different story for the crack propagation process of 3D-GLNN/Cu. Crack bridging by graphene/Cu interlocked structure (Fig. 6e) as well as 3D-GLNN (Fig. 6f and Supplementary Fig. 21) were captured in the stage (IV) and (V). The difference in the cracking morphology could be attributed to the different load transfer capabilities between 3D-GLNN/Cu and RGO/Cu. Benefited from the interlocking network structure, the 3D-GLNN bridged between the cracked matrixes postpones the extending progress of cracks[5,37]. To achieve effective interaction with cracks, the geometric size of the reinforcement should be large enough to match the crack size[39]. In this sense, 3D-GLNN interconnected by thousands of GLNs, enlarged the effective size to react with cracks.

We then applied MD simulations on two simplified models to study the effect of network architecture on the interfacial shear

stress. One single-layer graphene embedded in 13 layers of Cu on both sides was built to study the pull-out process of graphene in 2D-G/Cu (Fig. 6g). At the same time, a regular three-dimensional network composed of single-layer graphene with an all-$sp^2$ structure was built inside Cu matrix to study the interfacial behavior of 3D-G/Cu (Fig. 6h). The total numbers of the carbon atoms in the two models were controlled to be same. The pull-out force/energy-displacement curves of 2D-G/Cu and 3D-G/Cu were presented in Fig. 6i, j, respectively. A steady pull-out progress could be verified from the relatively smooth curves shape in 2D-G/Cu (Fig. 6i). The structure of 2D graphene was almost unchanged after the full pull-out from copper. While, things became much more complicated for the case of 3D-G/Cu (Supplementary Movie 5). According to the trend of total energy-displacement curve, the whole progress could be divided into three steps. The snapshots of the typical state A-D were presented in Fig. 6k. In the Stage 1 (0–4.1 Å), 3D-G was pulled out smoothly with little deformation which could be reflected by the linear increase of the total energy during this period. Besides, the copper matrix deformed elastically; In the Stage 2 (4.1 Å–12.7 Å), 3D-G was deformed by the resistance force of Cu matrix during the on-going pull-out displacement. The complex interaction between Cu and 3D-G induced energy fluctuation in the total energy and thus resulted in a violent rise and fall for the pull-out force. Plastic deformation could be observed in the copper matrix and the dislocation density increased sharply (Supplementary Fig. 24); In the Stage 3 (>12.7 Å), the 3D-G deformed into a planar structure and lost the effective interaction with the Cu matrix. The total energy of 3D-G decreased due to the instability of the deformed 3D-G model and resulted in a negative value of pull-out stress of this range. The on-going plastic deformation of copper matrix further increased the dislocation to a high level of $14.3 \times 10^{14}\,\mathrm{m}^{-2}$, suggesting a high dislocation strengthening capability of 3D-G. With the same displacement of 12.7 Å, the increased potential energy correlated to the work done by the pull-out force was measured as 164 eV for 3D-G/Cu model, which is one times more than that of 2D-G/Cu (79 eV). To compare the difference in the interfacial behaviors between 3D-G/Cu and 2D-G/Cu, it is important to convert the measured data into a unified physical quantity. Herein, we proposed to use the interfacial shear stress ($\tau$), which was estimated by the division of the average pull-out force ($F_{pm}$) with the projected area ($A$) in the X–Y plane given as[40]:

$$\tau = \frac{F_{pm}}{2A} = \frac{F_{pm}}{2wL} \tag{3}$$

where $w$ is the width of graphene (46.85 Å), $L$ is the projected length of graphene. The average pull-out force for 2D-G/Cu was determined as 10.6 nN and the corresponding interfacial shear strength was calculated as 69.7 MPa for the whole progress. Comparatively, the average pull-out force for the periods with increased total energy (Stage 1 and Stage 2) was estimated as 22.7 nN and the interfacial shear strength was obtained as 214.8 MPa, almost two times higher than that of 2D-G/Cu. If we focused on the steady pull-out stage (Stage 1), the pull-out force was measured as 28.5 nN and resulted in a high interfacial shear strength of 269.7 MPa, more than 2.8 times higher than that of 2D-G/Cu. Coincidently, both the theoretical and experimental results pointed to a similar three-times relationship between 3D-GLNN/Cu and 2D-RGO/Cu. The simulation results strongly verified that, compared with 2D graphene/Cu, the dramatically increased interfacial shear strength which contributes to a better load transfer is the main cause of the enhancement in mechanical strength for 3D graphene/Cu.

Based on the above results, the superior mechanical properties of 3D-GLNN/Cu could be attributed to the unique network architecture: First, the continuous 3D-GLNN restricted the growth of matrix grains and resulted in grain refinement strengthening mechanism; second, the network architecture brought thousands of

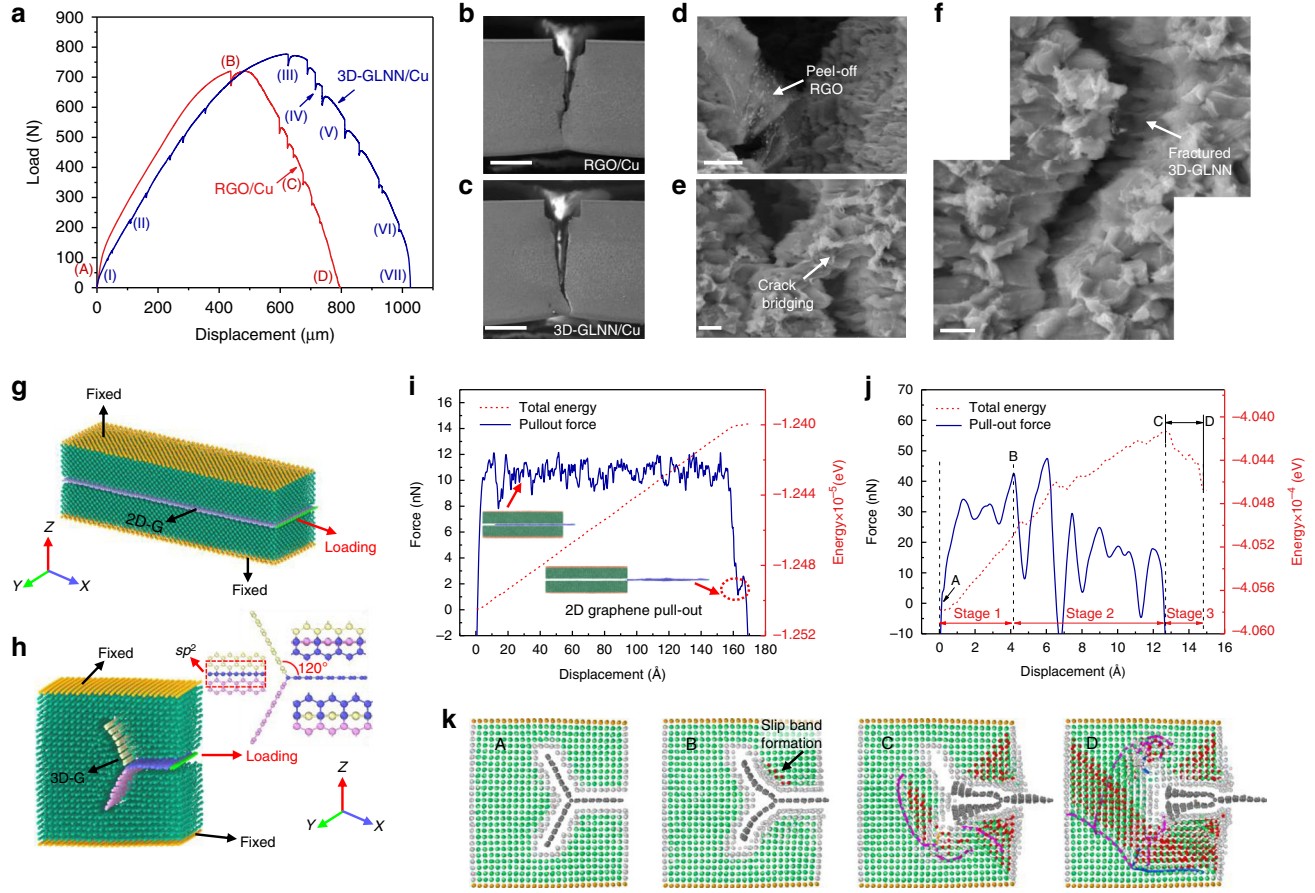

**Fig. 6 Comparison between 3D graphene/Cu and 2D graphene/Cu. a** Load-displacement curves of RGO/Cu and 3D-GLNN/Cu with several paused stages. **b, c** SEM images of tensile samples after in-situ tensile test of (**b**) RGO/Cu and (**c**) 3D-GLNN/Cu. Scale bar, 1 mm (**b, c**). **d** Typical SEM image of RGO peeled-off from the matrix in stage (C) during tensile deformation. Scale bar, 5 μm. **e, f** Typical SEM images of crack bridging by **e** graphene/Cu interlocked structure and **f** 3D-GLNN. Scale bar, 2 μm (**e, f**). **g, h** Atomic configurations in MD simulations of **g** 2D-G/Cu and **h** 3D-G/Cu, 3D-G was built with an all-*sp*² structure at an angle of 120 degree between three directions. **i, j** The simulated pull-out force-displacement curves of **i** 2D-G/Cu and **j** 3D-G/Cu, the 2D-G was pulled out steadily until fully separated from the matrix while 3D-G was deformed with a complicated progress which could be divided into 3 stages. **k** Typical snapshots corresponding to the point A-D during the deformation progress of 3D-G/Cu. 3D-G was blocked by the Cu matrix, the two wings of which were constrained parallel to the X–Y plane in D. The atoms in red indicated HCP transition formed on the {111} plane. The dislocation in magenta and blue are attributed to 1/6 <112> (Shockley) dislocation and 1/6 <110> (Stair-rod) dislocation, respectively.

interfaces in the matrix, which blocked dislocations from moving across the grain boundaries efficiently; Last but not the least, the 3D geometric feature of 3D-GLNN improved the interfacial shear stress and thus benefited a better load transfer strengthening mechanism. It should be stressed that the 2D graphene usually has a low interfacial shear stress in the composites due to the wrinkling of GNSs[41]. So the interfacial shear stress was much smaller than the theoretical estimation and the common interfacial failure mechanism for 2D graphene/metal is interfacial sliding followed by graphene pull-out, in which case graphene could not give full play to its outstanding mechanical properties[42]. The graphene network architecture, taking advantages of the interlocking behaviors between the matrix and the reinforcement, offers a new strategy to break through the limitations. From another point of view, the network architecture also enlarged the effective size of graphene into micrometer level, which equaled to the actual crack length and induced extrinsic toughening mechanisms such as crack bridging.

For the thermal transportation properties, the intrinsic TC of the reinforcement and extrinsic interface and architecture effect are both the key factors. Given the technical difficulty for accurate measurement of TC of 3D-GLNN in the copper matrix, herein an approximate value could be predicted according to the comparison with the reported work on graphene with the same defect

level. Based on the average $I_D/I_G$ ratio of about 0.75 (Supplementary Fig. 25), the density of defects in graphene, $N_D$, could be estimated by the following equation[43]:

$$N_D(cm^{-2}) = \frac{(1.85 \pm 0.5) \times 10^{22}}{\lambda^4}\left(\frac{I_D}{I_G}\right) \quad (4)$$

where $\lambda$ represents the laser wave length (532 nm). The $N_D$ value of 3D-GLNN is calculated to be $16.8 \pm 4.7 \times 10^{10}$ cm$^{-2}$. Thus the TC of 3D-GLNN could be predicted as about 450–500 W m$^{-1}$ K$^{-1}$ based on the experimental work by Malekpour et al.[44]. In view of the similar continuous and multi-layer feature of graphene, this value matches well with the reported data of graphene film/Cu system (445–538 W m$^{-1}$ K$^{-1}$)[45–47]. Restricted by the annealing temperature of 800 °C in the RTA step, the crystallinity and the intrinsic TC value of GLNs still have much room for improvement on the condition that an optimized GLNs growth procedures, such as plasma-enhanced chemical vapor deposition, could be achieved. Extrinsically, different from homogenous distribution and alignment architecture of GNSs, a continuous network structure may reduce both the Kapitza resistance and graphene-graphene contact resistance to a minimized level within the covalently bonded networks[48]. At the same time, the interconnected network structure of graphene/Cu provided extra electrons and phonons transfer

channels inside the matrix while maintaining the continuity of Cu. Moreover, the continuous graphene/Cu interfaces in the composites also reduced the electrons scattering in the boundaries, which benefits the good retention of high-temperature thermal conductivity. In contrast to the inevitable anisotropic thermal conductivity for the alignment or laminate architectures, 3D-GLNN/Cu did not compromise the thermal conductivity value in the perpendicular direction compared with that of the in-plane direction, achieving excellent near-isotropic transport properties.

In summary, taking advantage of the thermal stress induced by the mismatch of CTEs between the matrix and the reinforcement, novel continuous three-dimensional graphene-like network/copper composites with tightly-welded graphene structure were fabricated by a facile and scalable hot-pressing combined hot-rolling method. Both the experimental and MD simulation results strongly validated the important role of thermal stress in the effective welding between graphene layers thus constructing a regular and interconnected network architecture. Specially, for the first time the network architecture was verified to significantly improve the interfacial shear stress to about three times the value of two-dimensional graphene/Cu and thus promoted the load transfer strengthening. The idea of constructing a continuous graphene network in copper was proved to be not only effective in improving the mechanical properties (strength and toughness) but also beneficial to the enhancement of physical properties such as thermal and electrical properties of the metal matrix composites. Moreover, this strategy can also be extended to the preparation of other three-dimensional network composed by two-dimensional materials (h-BN, transition-metal sulphides or oxides) and their reinforced metal matrix composites (Cu, Mg, Al, Ti, etc.) for potential structural and functional applications.

## Methods

**Materials**. The Cu powders were purchased from Shanghai Yunfu Nanotechnology Co., Ltd. Sucrose was purchased from Tianjin Kemiou Chemical Reagent Co., Ltd. GO nanosheets were purchased from Chengdu Organic Chemicals Co. Ltd. All reagents were used without further purification.

**RTA growth of graphene-like nanosheets on Cu powders**. 0.160 g sucrose was first dissolved in the hybrid solution of ethanol and water (20 ml/40 ml) and stirred for 30 min to obtain a uniform and transparent solution. Afterward, 24.0 g Cu powders were added into the solution under stirring and then the mixture was sonicated for 20 min and then heated at 75 °C under constant magnetic stirring until the solution was vaporized completely. After drying at 80 °C for 4 h, the as-prepared sucrose/Cu precursor powders were put in a corundum boat and transferred to a CVD quartz tube. The quartz tube with sucrose/Cu precursor in the middle was put into the heating zone which was preheated to 800 °C and the annealing progress was maintained for 10 min under the Ar (200 ml/min) and $H_2$ (100 ml/min) atmosphere. Next, the powders were cooled down to room temperature by a rapid cooling treatment and then grinded to get GLNs/Cu composite powders.

**In-situ welding of GLNs into 3D-GLNN/Cu by hot-pressing**. Typically, the GLNs/Cu composite powders were placed in a graphite mold and then were consolidated into bulk composites with a pressure of 50 MPa at 800 °C for 60 min under vacuum of $10^{-4}$ Pa level. The heating rate of hot-pressing was 10 °C/min. During the hot-pressing progress, GLNs were welded into continuous 3D-GLNN in the Cu matrix. The consolidated 3D-GLNN/Cu bulk composites was 30 mm in diameter and 6 mm in thickness.

**Hot-rolled 3D-GLNN/Cu into densified bulk composites**. The hot-pressed 3D-GLNN/Cu composites were hot-rolled by a 70% reduction in thickness. The rolling reduction during each rolling cycle was 5% thickness.

**Fabrication of RGO/Cu bulk composites**. The RGO/Cu composites were fabricated as the contrast samples. Typically, 0.03 g GO was first dispersed in the hybrid solution of ethanol and water (20 ml/40 ml), followed by ultrasonication treatment for 2 h to obtain a dispersed suspension. Afterward, 24.0 g Cu powders were added into the solution under stirring and then then heated at 75 °C under constant magnetic stirring until the solution was vaporized completely. Next, the GO/Cu hybrid powders were dried and transferred to a quartz tube furnace. RGO/Cu

composite powders were obtained after reduction at 450 °C for 1 h under the Ar (200 ml min$^{-1}$) and $H_2$ (100 ml min$^{-1}$) atmosphere. Then, the RGO/Cu hybrid powders were fabricated into the final product with similar hot-pressed and hot-rolled processing steps.

**Characterization techniques**. Scanning electron microscopy (SEM) (Hitachi S-4800) and Transmission electron microscope (TEM) (JEOL JEM-2100F) were utilized to observe the microstructures of the powders and composites. Raman spectroscopy (Renishaw inVia Raman Microscope) with 532 nm Ar$^+$ laser was performed to characterize the graphene materials. X-ray photoelectron spectroscopic (XPS) measurement of the GLNs/Cu powders and 3D-GLNN/Cu bulk composites were carried out on a PHI 1600 ESCA system. The analysis of GLNs content in GLNs/Cu composite powders was performed by HCS-140 high frequency infrared carbon/sulphur determinator. Electron Backscattered Diffraction (EBSD) analysis was carried out by using a HKL Channel 5 system attached to SEM (FEI Nova NanoSEM 430) to evaluate the average grain size and the Taylor factor distribution of the composites. Focused ion beam (FIB) milling was used to perform FIB tomography (FEI Helios nanolab 600). The tomographic reconstruction was performed by using Avizo software.

**Mechanical properties measurements**. For static tensile testing, the obtained bulk samples were cut and polished to a dog-bone shape with a gauge length of 10 mm, a gauge width of 3 mm and a thickness of 2 mm. Tensile testing experiments were performed by a standard mechanical tester (Lloyd (AMETEK) EZ 20) with a crosshead speed of 0.5 mm min$^{-1}$ at room temperature. For in-situ SEM tensile test, the specimen machined with a small notch (the illustration is shown in Supplementary Fig. 18c) was fixed on a miniaturized deformation device system (Supplementary Fig. 18a) with a tensile speed of 2 μm s$^{-1}$ in a MAIA3 model 2016 SEM. The tested specimen was made by mechanical polishing and surface etching in a FeCl$_3$/HCl solution. The tensile progress was paused several times for observing the crack morphology in real time.

**Thermal properties measurements**. The laser flash method was carried out for the measurement of anisotropic thermal diffusion coefficient ($\alpha$) of the samples on a Netzsch LFA 467/447 Nanoflash for the in-plane and trough plane direction measurements. The specific heat ($C_p$) of the samples were measured on a NETZSCH DSC 214 differential scanning calorimeter (DSC) with a heating rate of 10 °C min$^{-1}$. The density ($\rho$) of the samples were measured by an electronic balance (Shimadzu AUY-120). The TC of the composites was obtained according to $K = \alpha \times \rho \times C_p$. Thermal expansion was measured with Netzsch dilatometer DIL 402C in Ar with heating and cooling ratios of 5 K min$^{-1}$ in the temperature range 35–550 °C. The size of the specimen was $25 \times 5 \times 3$ mm$^3$.

**Electrical conductivity measurements**. Electrical conductivity was tested by an FQR-7501 eddy-current electric conductivity instrument, and all the samples were wire-cut into a dimension of 10 mm × 10 mm × 2 mm and surface polished.

**Molecular dynamics simulations**. The large-scale Atomic/Molecular Massively Parallel Simulator (LAMMPS) were used for MD simulations[49]. Four different simulation models were used in this study, including a six-layer graphene (6LGs), six-layer graphene embedded in two single-crystal copper nanosheets (6LGs/Cu), single-layer graphene embedded in two single-crystal copper nanosheets (2D-G/Cu) and single-layer three-dimensional graphene embedded in single-crystal copper matrix (3D-G/Cu). The crystallographic orientation of Cu nanosheet is ⟨1 0 0⟩ and the lattice directions of [1 0 0], [0 1 0], and [0 0 1] are parallel with the X, Y, and Z coordinate axes, respectively. Graphene used for simulations had an armchair configuration. The interactions between carbon atoms were described by an adaptive intermolecular reactive empirical bond order (AIREBO potential) with a cut-off set as 1.95 Å[50], and that between Cu atoms were determined by the embedded atom method (EAM) potential[51]. A Lennard-Jones (LJ) potential was used to describe the Van der Waals interaction between C and Cu[40]. Before the MD simulation, initial configurations of graphene, copper and the composite models were relax for 100 ps at 300 K by using a Nosé-Hoover pressure barostat under the conditions of constant pressure and temperature (NPT) ensemble.

A 6LGs/Cu model with the size of 67.99 Å × 72.28 Å × 93.30 Å was designed to model the bonding condition between the graphene layers in two face-to-face-contact three-layer graphene/copper parts. The model 6LGs (67.99 Å × 72.28 Å × 16.75 Å) was set for comparisons. The surface of 6LGs in both models were created with randomly-distributed single-vacancy defects at a desired concentration. Periodic boundary condition was applied along the X-direction, while the boundary in the Y and Z direction were confined to eliminate rigid motion of the model during simulations. The equilibrium structures after relaxation at 293 K were heated to the high temperature (493 K, 693 K, 893 K, 1093 K, and 1293 K) step by step with each step for 30,000 timesteps under isothermal-isochoric (NVT) ensemble controlled by Nosé-Hoover thermostat. To evaluate the effect of welding, single-layer graphene in the middle was pull-out from the 6LGs with each pulling displace of 0.08 Å until it was completely pulled out from the embedded matrix controlled by microcanonical ensemble (NVE). Equilibrium was performed in each interval period of two continuous loading steps. The system was equilibrated by

using NVT ensemble for 10 ps to reach a new stable state. Then the total energy and pull-out force of the stable system was extracted for evaluating the pull-out force and the bonding behaviors between graphene layers.

To Model the pull-out behaviors of 3D-G/Cu, a single-layer three dimensional graphene was simplified into an all-$sp^2$ bonding network structure (as shown in inset of Fig. 6h). The model size was designed as $54.30 \text{ Å} \times 46.85 \text{ Å} \times 50.61 \text{ Å}$ for 3D-G/Cu and $162.32 \text{ Å} \times 46.85 \text{ Å} \times 50.61 \text{ Å}$ for 2D-G/Cu. The sum of the graphene length of 3D-G in three directions equals to that of the 2D-G model with the same numbers of atoms. Discrete displacements were imposed on the carbon atoms on the right end of graphene, with each pulling step of 0.08 Å until it was completely pulled out from the embedded matrix controlled by NVE ensemble. The two top layers and one bottom Cu layer in the Z-direction was fixed during the whole pull-out progress. After each displacement loading step, the system was equilibrated using NVT ensemble for 10 ps to reach a new stable state. Then the total energy and pull-out force were extracted for evaluating the pull-out force interfacial shear stress between graphene and copper.

## Data availability

The data that support the findings of this study are available from the corresponding authors upon reasonable request. The source data underlying Fig. 4e are provided as a Source data file.

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

## Acknowledgements

We gratefully acknowledge the financial support by the National Natural Science Foundation of China (Grant No. 51771130, 51531004, 51675211, and 51422104), the Tianjin youth talent support program, the Tianjin Natural Science Funds for Distinguished Young Scholar (Grant No. 17JCJQJC44300) and the Tianjin Science and Technology Support Project (Grant No. 17ZXCLGX00060).

## Author contributions

X.Z., C.N.H., and N.Q.Z. conceived the idea and designed the experiments. X.Z. performed the experiments. F.L Z., Y.X.X., and M.C.M. built the simulation models of 2D-G/Cu and 3D-G and conducted the MD simulation. E.Z.L., C.S.S., and D.L. analyzed the simulation results. C.N.H. and X.Z. analyzed the data and wrote the paper. C.N.H. and N.Q.Z. supervised the whole work. All authors discussed the results and commented on the paper.

## Competing interests

The authors declare no competing interests.
