## [Peer Review File · Nature Communications]

Reviewers' comments:

Reviewer #1 (Remarks to the Author):

Title: Metal-thermal-stress-induced fabrication of 3D graphene-like network for reinforcing copper matrix composites with superior properties

The paper is presenting a method of producing 3D carbon-copper composites, and reports on the mechanical and physical properties of the products. The paper is well-presented. However, there are some minor and some key questions as follow:

Minor points:

Introduction

1. "From learning the... traditional monolithic materials"

The long and general explanations like that pointed above with several references are not appropriate.

2. "Besides, the high processing temperature which exceeds the melting point of Cu (1083 °C) could introduce massive structural defects to the skeleton of 3D graphene preform." This statement doesn't seem to be correct. Is there any evidence for this?

Key points

1. In the process described in the paper, sucrose is coated on Cu particles, and carbonized at 800°C. This process is different from CVD, although a CVD-type reactor is used.

2. Fig. 2d (high resolution TEM of carbons) and 3f (Raman data) are in disagreement. It seems that sucrose has been carbonized into amorphous carbon, which is obviously far from graphene.

3. Considering the poor nature of the carbon layers (Fig. 3f), it is extraordinary difficult that this carbon material can have greater thermal conductivity values than copper. Therefore, Figure 5h, which shows that the thermal conductivity of the C-Cu composite is greater than that of Cu, should be revisited. Also, sufficient evidence to support the claim should be provided. It is suggested that the authors check the thermal (electrical) conductivity of the carbon material alone.

Reviewer #2 (Remarks to the Author):

In this manuscript, the authors developed a general way to construct a 3D-CGN/metal composite. The strategy is based on the powder-metallurgy growing the graphene nanosheets on the surface of copper powders and welding graphene nanosheets by thermal-stress. The constructed 3D-CGN/copper composite showed improved mechanical properties, thermal and electrical properties. The results are interesting and important, especially this strategy can be applied to the preparation of other 3D networks with good physical properties/metal matrix composites, and I think it can be published after the authors address the following issues:

1) Page 8: "For the first step, a uniform coating of GLNs was in-situ grown on the Cu powder surface by a typical CVD process at 800 °C, at which temperature the sintering between Cu powders just started." Is this CVD process at 800 °C related to the size of Cu powder? or is the uniform coating of GLNs obtained on more large Cu powder surface and is it temperature-dependent?

2) Page 16: "It is no doubt that the enormous thermal stress which is dozens of times higher than the external press applied could cause a big impact on the embedded GNSs." The question is when and how the enormous thermal stress is released since it is dozens of times higher than the external press? We know the boundary is fixed in the MD simulation, so the Z-direction stress of Y-Z single-layer slice in the middle of 6LGs/Cu is increased at 1093 K compared to 493K. Does the thermal stress fracture the GLN coating and then the thermal stress is released?

3) What is the temperature for the tensile response of the pure Cu matrix, 3D-GLNN/Cu and RGO/Cu composites? Why do they have similar curve but different fracture stress?

4) The authors mentioned that 3D-GLNN/Cu bulk composite has an interpenetrated network, suggesting the electron heat transfer of copper remains intact and still acts as the dominant role in deciding the thermal conductivity of the composites at room-temperature. Since the 3D-GLNN provides phonon transfer channels as well, how do you know it is electron heat transfer is the dominant thermal conduction in the composites?

5) Page 40: "using a Nosé-Hoover pressure barostat under the conditions of constant pressure and temperature (NPT) ensemble." What is the Nosé-Hoover pressure barostat?

Responses to the Reviewers' Comments

Reviewer #1 (Remark to the Author):

The paper is presenting a method of producing 3D carbon-copper composites, and reports on the mechanical and physical properties of the products. The paper is well-presented.

However, there are some minor and some key questions as follow:

Minor points:

Introduction

1. "From learning the... traditional monolithic materials"

The long and general explanations like that pointed above with several references are not appropriate.

Response: We thank the reviewer very much for the kind comments.

We have removed the explanations from the Introduction part as it is not relevant to the main text.

2. "Besides, the high processing temperature which exceeds the melting point of Cu (1083 °C) could introduce massive structural defects to the skeleton of 3D graphene preform." This statement doesn't seem to be correct. Is there any evidence for this?

Response: We thank the reviewer very much for the kind comments.

As we have mentioned in the manuscript, the freestanding 3D graphene scaffolds were first prepared by self-assembling 2D-GNSs through freeze-drying or direct growing on the 3D metal templates by Chemical vapor deposition (CVD) method. For the former, the 3D graphene network was constructed by GNSs (commonly GO nanosheets) as building units via C-X bonds (X=C, O, S atoms, etc.)^{R1}. Taking the widely-used hydrothermal method as an example, the interconnected feature of 3D graphene network was achieved through the cross-linking of the functional groups such as OH, COOH, and epoxy groups located on GO nanosheets edges. These 3D graphene aerogels or foams contain massive oxygen-based functional groups (OBFGs) in their skeletons^{R2}. Considering the high temperature applied to melt copper (>1250 °C), a typical annealing process could happen on GO nanosheets during the

mixing procedure with copper melt, during which period the OBFs were decomposed into H₂O and CO₂ followed by a gradual structural conversion from sp³ lattice to typical sp² lattice of GO nanosheets. Limited by the temperature, however, the driving force of sp³ to sp² conversion is relatively weak thus leading to a defected structure of graphene with vacancies and residual OBFs on the surface^{R3}. This phenomenon could be verified by the increase of intensity ratios between D band and G band (I_D/I_G ratios) for the annealed 3D graphene network at 1000~1500 °C compared with that of the raw samples from the Raman spectra^{R4}. At the meantime, the trace amount of oxygen in the copper ingot could be released in the melting system and reacted with dangling-bond edge carbon atoms on the surface of 3D graphene network, which could also cause structural damage as well^{R5}.

Based on the above , both the intrinsic structural factors and extrinsic processing factors could introduce defects to the skeleton of 3D graphene network in the melting-related processing techniques.

Reference R4 and R5 have been added in the revised manuscript.

Key points

1. In the process described in the paper, sucrose is coated on Cu particles, and carbonized at 800°C. This process is different from CVD, although a CVD-type reactor is used.

Response: We thank the reviewer very much for the kind comments.

We agree with the reviewer that in the present work the synthesis process of graphene-like nanosheets (GLNs) is different from the common CVD method in the usual sense. Referenced from the published papers, the copper catalyzed CVD growth of graphene materials follows the surface-mediated self-limited growth mechanism which could be briefly explained as below^{R6,R7}: First, gaseous hydrocarbons (or liquid/solid carbon source decomposed into gaseous hydrocarbons) were brought and adsorbed onto the copper foil surface by carrier gas and dissociated into active carbon species under the catalysis of the copper substrate. Second, the active carbon species

are aggregated into small graphene nuclei and grow into large lateral size with sufficient attachment of newly formed carbon species. Finally, a continuous graphene film forms when adjacent graphene domains merge together. Sun et al. first reported the successful CVD-synthesis of monolayer graphene on the Cu foil by using sucrose as solid carbon source^{R8}. In this work several differences could be identified compared with Sun et al's method: 1) **Different types of substrate and supporting methods for solid carbon source.** In Sun et al's work, they loaded sucrose powders directly on the surface of a 25 μm thick Cu foil as solid feed stock. While, in the present study we used spherical Cu powders with an average diameter of less than 2 μm as the template and also the catalyst. In view of the relatively small surface area of the individual sphere powder and the packing characteristics of Cu powders, the combined solution-mixing and drying methods was used to coat a thin layer of sucrose on the surface of Cu powders. So it is believed that the subsequent pyrolysis of carbon source process took place in a more confined space between Cu powders in comparisons with the vast space above Cu foil for gas flow of the dissociated hydrocarbons in Sun et al's work.

2) **Different growth mechanisms of graphene (or GLNs).** Currently, a consensus has been made on the CVD growth mechanism of graphene using solid carbon source from Sun's work and also other reported papers^{R8,R9,R10}. It is widely recognized that for carbon sources without aromatic moieties such as polymethylmethacrylate (PMMA) and sucrose, the steps of decomposition into gaseous hydrocarbons and the gas re-adsorption are of the initial importance which decide whether the graphene nucleation on Cu foil could take place^{R10}. In this work, the sucrose/Cu precursor powders sent to the pre-heated hot-zone in a tube furnace went through a two-step pyrolysis and GLNs growth process. During the first step, the sucrose coating pyrolyzed into amorphous carbon and H_2O , the latter escaping from the powder surface in gas state with the rapid heat treatment. In the second step, the amorphous carbon rapidly react with H_2 to form hydrocarbons (or reactive intermediates) both in the gas and liquid states^{R9}. The gaseous hydrocarbons were then partly adsorbed on Cu surface and dissociated into active carbon species. The subsequent graphene

nucleation and domain connection were similar to those in the CVD process. While the liquid-state hydrocarbon decomposed and annealed to obtain defected graphite structures, which inevitably contains parts of amorphous carbon due to the low annealing temperature of 800°C. The formation of gaseous hydrocarbons could be indirectly verified by the severe loss of the carbon content of 66% in the GLNs/Cu powders compared with the nominal carbon content in the sucrose/Cu precursors.

Based on the above analysis, we have changed the name of the second step of the fabrication processes “CVD” in the previous version into “rapid thermal annealing” for accuracy. Besides, we have also revised the corresponding description and discussion of GLNs growth in the manuscript.

2. Fig. 2d (high resolution TEM of carbons) and 3f (Raman data) are in disagreement. It seems that sucrose has been carbonized into amorphous carbon, which is obviously far from graphene.

Response: We thank the reviewer very much for the comments.

- 1) As discussed in the response of Question 1, we reckon the GLNs and 3D-GLNN as patched graphene, which could be understood as carbon material between ideal graphene and amorphous carbon^{R11}. In order to prove this point, the contour of I_D/I_G ratio in 3D-GLNN from the corresponding area in Figure 3g was plotted. It could be calculated from Figure R1 that the mean value of I_D/I_G was about 0.75, suggesting a relatively high level of defects in 3D-GLNN. The defects could be attributed to both the intrinsic disorder in the GLNs grown on Cu powders from RTA process and the structural damage from extrinsic densification and surface etching. Overall, the 3D-GLNN could be the category of nanocrystalline graphene with small ratio of amorphous carbon^{R12}. Furthermore, we have also supplemented the HRTEM images of GLNs which were obtained after etching Cu powders. It is verified that the edge of GLNs could be clearly distinguished, most of GLNs demonstrate mono-, double- or multi-layer structures (Figure R1a, c, and d). From figure R1b, both the graphene domains (with good crystallinity) and amorphous carbon area could be observed. The primary graphene nanodomains,

of which the average size were measured as about 5 nm with irregular shapes and distinguish boundaries, were interconnected to form the “patched graphene” structure.

- 2) Owing to the fact that the HRTEM image in Figure 2d and Raman spectrum presented in Figure 3f have different dimensions, in our opinion they are not contrary to each other. In the present work, the laser spot size for Raman spectra measurement of GLNs and 3D-GLNN was $\sim 1 \mu\text{m}^2$, which is 2~3 orders larger than the view field of HRTEM characterization^{R13}. The Raman spectra present an averaged result from hundreds of graphene nanodomains as well as amorphous carbon in GLNs and 3D-GLNNs. While the HRTEM in Figure 2d gave a typical morphology of an edge area in GLNs within a limited size of less than 100 nm^2 . In the manuscript, we have mentioned that the in-situ synthesized carbon materials on the Cu powders are “graphene-like nanosheets (GLNs)” which suggest that these materials are not the so-called graphene in the strict sense. Actually, we are trying to build our own modified rotating plasma enhanced chemical vapor deposition (PECVD) system and it is expected that the quality of GLNs will be improved in the future.

Figures R1 and R2 and the related discussions have been added in the revised manuscript.

Figure R1. Contour of the intensity ratio of the D band to the G band (I_D/I_G) for 3D-GLNN in a $10 \mu\text{m} \times 10 \mu\text{m}$ area.

Figure R2. HRTEM images of GLNs showing (a) edge area with thin-thickness feature, (b) graphene domains and amorphous carbon on the surface, (c, d) edges of different graphene layers.

3. Considering the poor nature of the carbon layers (Fig. 3f), it is extraordinary difficult that this carbon material can have greater thermal conductivity values than copper. Therefore, Figure 5h, which shows that the thermal conductivity of the C-Cu composite is greater than that of Cu, should be revisited. Also, sufficient evidence to support the claim should be provided. It is suggested that the authors check the thermal (electrical) conductivity of the carbon material alone.

Response: We thank the reviewer very much for the comments and suggestions.

We understand the reviewer's concern about the thermal conductivity (TC) of the 3D-GLNs/Cu as there are few papers indicating the positive effect of graphene addition on the thermal conductivity of copper and other metal matrix (Al, Mg, etc.). Unlike the nonmetal matrices (polymers, ceramics), the metal matrices themselves are good conducting materials. So the factors associated with the enhancement of the TC in MMCs could be much more complex besides the intrinsic TC value of reinforcement materials. Here we explain the enhancement of TC in the 3D-GLNN/Cu composites from two aspects:

1) **The intrinsic thermal conductivity of graphene reinforcement.**

a) First of all, it is important to notice that the Raman spectrum can not reflect the intrinsic thermal/electrical conductivity (EC) completely. Below we listed some of the reported data on the graphene or graphene-like materials synthesized by annealing method using solid carbon source (or as sacrifice template such as C_3N_4) (Table R1). It can be inferred from the FWHM(G) and I_D/I_G ratio data that most of them consist of defects such as vacancies and residue oxygen based functional groups. However, the measured TC/EC values of these carbon materials were high, some of which even equal to those of the CVD derived few-layer graphene materials using gaseous carbon source^{R14,R15}. From another aspect, Zhao et al. verified the relation between defects level of mono-layer graphene and its TC value^{R16}. Their research results suggested that oxygen groups such as carbonyl pairs could be the main reason of the TC reduction compared with other types of defects such as vacancies and hydroxyl and epoxy groups. Given the fact that the graphene/graphene-like materials obtain from RTA method usually contains negligible amount of oxygen after the high temperature annealing process, which could partially explained their high TC value.

b) It is well known that, the electrical/thermal conductivity measurement method for nanomaterials, especially graphene, has been controversial due to limitation of the nanoscale size effect and also the instrumental error. In the present work, the GLNs synthesized on Cu powders (0.5~2 μm in diameter) have a relatively small surface area and a curved surface morphology. For the normally-used confocal micro-Raman spectroscopy method^{R17}, it is a significant challenge for the GLNs to be etched and transferred onto a substrate and measured as a flat graphene film. To measure the conductivities of powder based 3D-GLNs and 3D-GLNN, it is necessary to etch the matrix Cu powders or Cu bulk materials with etching agents like $FeCl_3/HCl$ suspension, in which process impurities such as Cu_2O and residual $FeCl_3$ were

inevitably introduced to the Raman samples. Moreover, the subsequent powder compacting step could cause structural damage to the 3D carbon skeleton and causes the formation of numerous numbers of graphene-graphene interface^{R18}. Therefore, the measured value could be far below the intrinsic electrical/thermal conductivity of 3D-GLNs and 3D-GLNNs. In addition, it is also controversial to compare the measured EC/TC value of the reinforcement to directly compare with those of the bulk materials measured by different methods. To the best of our knowledge, the precise measurement of EC/TC value of the carbon materials reinforced MMCs was rarely reported in previous publications. Based on the average I_D/I_G ratio of about 0.75 (Figure R1), the density of defects in graphene, N_D , could be estimated by the following equation^{R19}:

$$N_D(cm^{-2}) = \frac{(1.85 \pm 0.5) \times 10^{22}}{\lambda^4} \left(\frac{I_D}{I_G} \right) \quad (1)$$

Where λ represents the laser wave length (532 nm). The N_D value of 3D-GLNN is calculated to be $16.8 \times 10^{10} cm^{-2}$. Thus the TC of 3D-GLNN could be predicted as about 450~500 $W m^{-1}K^{-1}$ based on the experimental work by Malekpour et al^{R20}. This value matches well with the reported data of multi-layer graphene film/Cu system (445~538 $W/m^{-1}K^{-1}$)^{R21,R22,R23}. Considering the continuous feature of 3D-GLNN in the Cu matrix, we believe the calculated value above is reasonable.

- 2) **The extrinsic interface and architecture factors.** Usually, interface plays a key role in deciding the final properties of MMCs. For example, in the system of graphene reinforced copper matrix composites with homogenous distribution, the interfacial Kapitza resistance is the main factor limiting the effective TC of the graphene interlayer and thus leads to a reduced TC of the whole composites^{R24}. The situation changed when Chu et al.^{R25} reported a much enhanced TC value of 525 $W m^{-1}K^{-1}$ (50% higher than that of pure Cu) of GNS network/Cu composite which was realized by the optimized graphene size and good alignment with a

high content of 35 vol.%. In their work, the Kapitza resistance was not the main resistance considering the accelerating upward trend of TC with the increase of GNS volume fraction. Instead, the graphene-graphene contact resistance becomes the dominant role. In the present work, an interpenetrating network was confirmed by the FIB-3D reconstruction results of the 3D-GLNN/Cu (Figure 2h). Different from homogenous distribution and alignment architecture of GNSs, a continuous network structure may reduce both the Kapitza resistance and graphene-graphene contact resistance to a minimum level within the covalently bonded networks. Meanwhile, at the nanoscale, Melta et al. verified that graphene coating around Cu nanowires led to an enhanced elastic surface scattering of electrons thus building a confined space for electron transportation^{R26}. We propose this mechanism could also work well and facilitate the electron to transport in the Cu matrix encapsulated by 3D-GLNN considering the interpenetrating network feature and reduced the grain boundary scattering (as illustrated in the inset of Figure 5j), which is believed to be the main reason for the reduction of TC at high temperatures. By comparing the TC values at different temperatures between 3D-GLNNs/Cu and RGO/Cu, the gradual TC reduction trend with the increase of temperature could be good evidence to support the success formation of an interpenetrating network structure. While in RGO/Cu, the discontinuous distribution of RGO nanosheets failed to take effect and resulted in a similar quick reduction trend with the increase of temperature.

To sum up, based on both the calculated thermal conductivity of 3D-GLNN ($\sim 450 \text{ W m}^{-1}\text{K}^{-1}$) and its continuous network architecture effect, the enhancement of thermal conductivity of 3D-GLNN/Cu compared with pure Cu at room temperature was reasonable despite of the incomplete theory of the thermal conductivity of composites with interpenetrating network.

The related discussions have been given in the revised manuscript.

Table R1. The summarized reported data on the graphene or graphene-like materials synthesized by annealing method using solid carbon source.

Name	Preparation method	FWHM (G)	I _D /I _G	TC (W/m·K)	EC (S/m)	Reference
modified-graphene film	(GO+glucose) annealing at 1000°C (Ar)	77.9	1.03	1300	–	Li, J Mater Sci, 2019 ^{R27}
N doped graphene film/Cu	(Silk fibroin+Cu) microwave plasma treatment	115.3	0.85	538	6.25×10 ⁴	Zheng, Sci Rep, 2018 ^{R22}
C/Cu film	(Lignin+Cu) annealing at 1000°C (Ar)	73.0	0.23	478	–	Luo, Nanomaterials, 2019 ^{R23}
Graphene powders	(Glucose+FeCl ₃) annealing at 700°C (Ar)	34	0.35	--	768 (powder)	Zhang, Chem Sci, 2014 ^{R14}
3D strutted graphene	(Glucose+NH ₄ Cl) annealing at 1350°C (Ar)	35.8	1.19	--	20,000	Wang, Nat. Commn. 2013 ^{R15}
nitrogen-rich graphene-like carbon	(g-C ₃ N ₄ +CH ₄) deposition at 950°C (Ar)	103.7	0.99	--	693	Yang, Carbon, 2018 ^{R28}
p-doped graphene	(FeCl ₃ +styrene-co-methacrylic acid on PS template) annealing at 1000°C(H ₂)	44.6	0.46	--	65,000	Lee, ACS Nano, 2013 ^{R29}
3D graphene network	(PVA+FeCl ₃ on SiO ₂ template) annealing at 1000°C (Ar, H ₂)	26.0	0.15	--	5,200 (powder)	Yoon, Sci Rep, 2013 ^{R18}

Reviewer #2: In this manuscript, the authors developed a general way to construct a 3D-CGN/metal composite. The strategy is based on the powder-metallurgy growing the graphene nanosheets on the surface of copper powders and welding graphene nanosheets by thermal-stress. The constructed 3D-CGN/copper composite showed improved mechanical properties, thermal and electrical properties. The results are interesting and important, especially this strategy can be applied to the preparation of other 3D networks with good physical properties/metal matrix composites, and I think it can be published after the authors address the following issues:

1. Page 8: "For the first step, a uniform coating of GLNs was in-situ grown on the Cu powder surface by a typical CVD process at 800°C, at which temperature the sintering between Cu powders just started." Is this CVD process at 800°C related to the size of Cu powder? or is the uniform coating of GLNs obtained on more large Cu powder surface and is it temperature-dependent?

Response: We thank the reviewer very much for kind comments.

As we have shown in Supplementary Figure 1 that the annealing temperature of 800°C was chosen to maintain the powder morphology of GLNs/Cu and avoid serious sintering between Cu powders (at a temperature above 900°C). It is well known that the driving force of sintering for the stacking powders with smaller radius is larger than that of one with bigger powder size ($\Delta P = 2\gamma/r$, where ΔP is the pressure difference induced by surface tension, γ and r are the surface tension and radius of spherical powders). Therefore, it is reasonable that if Cu powders with even smaller radius compared with those used in this work, the annealing temperature should be even lower than 800°C to avoid serious sintering between the stacking powders. Experimentally, we have also used different Cu powders, namely irregular Cu powders with an average of size of 5 μm (5R-Cu) and spherical Cu powders with an average diameter of 40 μm (40S-Cu), as the template and catalyst to synthesize GLNs on their surface under the same preparation conditions. Some of the 5R-Cu

powders were partially sintered together (Figure R3a, b), while only isolated powder morphology was observed for GLNs/40S-Cu powders after annealing at 800°C (Figure R3d). And it turned out that for both cases GLNs were grown homogeneously on the Cu powder surface (Figure R3b, e and f). To further elucidate the influence of different Cu powders morphologies on the formation of 3D-GLNN, we prepared two groups of hot-pressed samples with 5R-Cu and 40S-Cu powders, respectively. It could be seen from Figure R4a that an incomplete and irregular network structure is exposed after surface etching. The magnified SEM images in Figure R4b, c validate that some of the GLNs were damaged after hot-pressing which may be attributed to the non-uniform thermal stress caused by the irregular shape of Cu powders. While for GLNs/40S-Cu-HP, despite of the successful formation of a continuous network structure in Cu matrix, numerous voids nucleated in the junction area (as shown in Figure R4d-f). It is suggested that the 40S-Cu powders are fully encapsulated by the continuous graphene, thus hindering from forming effective sintering necks during hot-pressing. As discussed above, the successful construction of 3D-GLNN/Cu requires the Cu powders in regular spherical shape and have relatively small size to facilitate partial sintering between Cu powders during GLNs synthesis and thus form the interpenetrating network structure afterwards. The annealing temperature at 800°C was the optimized choice and we suggest that the annealing temperature should be selected carefully in order to control the GLNs/Cu powder morphology required for the successful construction of interpenetrating network in the bulk composites.

Figures R3 and R4 and the related discussions have been given in the revised manuscript.

Figure R3. SEM images of GLNs/Cu hybrid powders fabricated by (a-c) 5 μm irregular Cu powders and (d-f) 40 μm spherical Cu powders.

Figure R4. SEM images of the 3D-GLNNs exposed from the etched surface in 3D-GLNNs/Cu: (a-c) GLNs/5R-Cu-HP, (d-f) GLNs/40S-HP

2. Page 16: "It is no doubt that the enormous thermal stress which is dozens of times higher than the external press applied could cause a big impact on the embedded GNSs." The question is when and how the enormous thermal stress is released since it is dozens of times higher than the external press? We know the boundary is fixed in the MD simulation, so the Z-direction stress of Y-Z single-layer slice in

the middle of 6LGs/Cu is increased at 1093 K compared to 493K. Does the thermal stress fracture the GLN coating and then the thermal stress is released?

Response: We thank the reviewer very much for kind comments and questions.

The damage of the GLN layer is related to temperature and boundary conditions. In the model of the manuscript, we set a fixed boundary in order to obtain a stable thermal stress. The fixed boundary limits the degree of freedom in the YZ direction of graphene. The simulation results show that graphene is not damaged, and the value of thermal stress could be measured.

In fact, we also performed pre-simulations where the graphene layer was not displacement-constrained. Fracture occurred in the unconstrained graphene layer when the temperature increased to 693 K (Figure R5), and the thermal stress was released at the mean time. For the case that graphene is fractured under high thermal stress without constraint, the system cannot maintain the relaxation process and thus the temperature increases irreversibly as observed at 693 K (Figure R6).

The results from pre-simulations showed that thermal stress is released through deformation and the failure of graphene without constrains. To determine the thermal stress, we fixed the boundaries in the manuscript.

Figure R5. Pre-simulations. Upper-graphene and the lower-graphene was fixed, and the copper substrate behind was constrained at bottom. (Temperature at 693 K, Time: left 88 ps, right 92 ps)

Figure R6. Temperature-time curves for the pre-simulation results.

3. What is the temperature for the tensile response of the pure Cu matrix, 3D-GLNN/Cu and RGO/Cu composites? Why do they have similar curve but different fracture stress?

Response: We thank the reviewer very much for the questions.

The tensile tests of pure Cu matrix, 3D-GLNN/Cu and RGO/Cu were carried out at room temperature. The engineering stress-strain curve of 3D-GLNN/Cu in Figure R7 were divided into four distinct regimes (I-IV) by short dash lines as a convenience to readers: (I) elastic loading up to the yield point ($\sim 0.2\%$); (II) strain hardening from the yield point to about 2% strain, in which process the dislocation interactions occurred and led to dislocation pile-up at the Cu-graphene interface; (III) steady flow at a nearly constant stress which indicates the dislocation in regime II may be quickly balanced by dislocation annihilation at the 3D-GLNN/Cu interface; (IV) elongation after peak stress until final fracture, which is dominated by the interaction between 3D-GLNN and the micro-cracks. By comparison, the stress curve of pure Cu displayed a different feature. It contains only three typical process without regime III. Inversely, a much larger regime II indicates that the tensile behavior of pure Cu reflects a typical strain-hardening feature of recrystallized Cu with coarse grain sizes (as shown in Supplementary Figure 11b and c). After reaching the peak stress, the gauge area reduced quickly due to the necking-related failure mechanism in regime IV. For RGO/Cu, it only has two typical regimes of I and II. In view of the relatively

weak restriction of grain boundary movement in RGO/Cu with 2D RGO nanosheets, an obvious recrystallization also occurred and resulted in a relatively low dislocation density in the hot-rolled samples. So the curve trend in regime II for RGO/Cu was similar to that of pure Cu. It should be noticed that the inevitable agglomeration of RGO nanosheets and also the weak RGO/Cu interface strength induced the formation of massive micro-cracks in the bulk composites after the initiation of plastic deformation. A pre-fracture occurred as a result of quick crack propagation.

Figure R7. Tensile stress-strain curves of pure Cu, RGO/Cu and 3D-GLNN/Cu. The upside-down triangles in blue indicates the peak stress point in the curves of RGO/Cu and pure Cu, respectively.

Based on the above analysis, it could be concluded that the ultimate tensile strength (UTS) (peak stress) was affected by a combined effect of initial-state grain structure as well as the specific plastic deformation behavior. The high UTS and fracture elongation of 3D-GLNN/Cu could be explained as below: First, the formation of a continuous network structure significantly increased the yield strength by the synergistic strengthening mechanisms of grain refinement, load transfer and geometrically necessary dislocations storage (as detailly discussed in the Supplementary Information). Second, 3D-GLNN located at the grain boundaries block the dislocation and led to a gradual strain hardening of the matrix followed by a dynamically-equilibrated process of dislocation multiplication and annihilation until

reaching the peak stress. Thanks to the uniform network structure, the plastic deformation was found to be more homogenous than RGO/Cu. Finally, the main cracks initiated and they were postponed to pass through the whole cross section by the effective crack bridging effect of 3D-GLNN. In this way, the stress reduced with the slow shrink of the gauge area until reaching the final fracture of the whole composites.

4. The authors mentioned that 3D-GLNN/Cu bulk composite has an interpenetrated network, suggesting the electron heat transfer of copper remains intact and still acts as the dominant role in deciding the thermal conductivity of the composites at room-temperature. Since the 3D-GLNN provides phonon transfer channels as well, how do you know it is electron heat transfer is the dominant thermal conduction in the composites?

Response: We thank the reviewer very much for the questions.

We agree with the reviewer's opinion that both the electron transfer mechanism in the continuous Cu matrix and the phonon transfer mechanism could be effective in thermal transfer in 3D-GLNN/Cu bulk materials. In view of the relatively small weight percentage measured as 0.096 wt. % of 3D-GLNN and the two-dimensional geometric size of its building blocks embedded between Cu grains, Cu matrix occupies most of the volume in the bulk composites. To roughly compare the contribution of the two mechanisms mentioned above, we apply to a simple prediction method following the electron-transfer-related Wiedemann-Franz law $K / \sigma = LT$, where σ is the EC and $L = (\pi^2 / 3)(k_B / q)^2 \approx 2.44 \times 10^{-8} W\Omega K^{-2}$ is the Lorenz number^{R30,R31}. The TC value of the composites based on electron heat transfer (K_e) could be predicted by $K_e = (\sigma_c K_m / \sigma_m)$. In this way, K_e was predicted as $387 Wm^{-1}K^{-1}$, which accounts for ~94% of the TC measured by laser flash method ($413 Wm^{-1}K^{-1}$). So this result roughly estimated that electron heat transfer is the main mechanism.

The related discussions have been given in the revised manuscript.

5. Page 40: "using a Nosé-Hoover pressure barostat under the conditions of constant pressure and temperature (NPT) ensemble." What is the Nosé-Hoover pressure barostat?

Response: We thank the reviewer very much for the question.

This is a time integration on Nosé-Hoover style non-Hamiltonian equations, which are designed to generate positions and velocities sampled from the canonical (NVT), isothermal-isobaric (NPT), and isenthalpic (NPH) ensembles. The thermostating and barostatting are achieved by adding some dynamic variables which are coupled to the particle velocities (thermostating) and simulation domain dimensions (barostatting). The barostat can be coupled to the overall box volume, or to individual dimensions, including the xy, xz and yz tilt dimensions. The external pressure of the barostat can be specified as either a scalar pressure (isobaric ensemble) or as components of a symmetric stress tensor (constant stress ensemble).

The equations used are from Shinoda et al.^{R32}, which combine the hydrostatic equations of Martyna, Tobias and Klein^{R33} with the strain energy proposed by Parrinello and Rahman^{R34}.

In summary, it is a widely used temperature/pressure control method, which can update the atomic velocity and position in each step, and is widely used in the MD simulations.

References:

- R1. Chen, W. et al. Polymeric graphene bulk materials with a 3D cross-linked monolithic graphene network. *Adv.Mater.* **31**, 1802403 (2019).
- R2. Xu, Y., Sheng, K., Li, C. & Shi, G. Self-assembled graphene hydrogel via a one-step hydrothermal process. *ACS Nano* **4**, 4324-4330 (2010).
- R3. Botas, C. et al. Critical temperatures in the synthesis of graphene-like materials by thermal exfoliation–reduction of graphite oxide. *Carbon* **52**, 476-485 (2013).
- R4. Chen, M. et al. Annealing temperature-dependent terahertz thermal – electrical conversion characteristics of three-dimensional microporous graphene. *ACS*

Appl. Mater. Inter. **11**, 6411-6420 (2019).

R5. Bydałek, A.W. Role of carbon in the melting copper processes. *Archives of Foundry Engineering* **11**, 37-42 (2011).

R6. Li, X., Cai, W., Colombo, L. & Ruoff, R.S. Evolution of graphene growth on Ni and Cu by carbon isotope labeling. *Nano Lett.* **9**, 4268-4272 (2009).

R7. Li, Z. et al. Low-temperature growth of graphene by chemical vapor deposition using solid and liquid carbon sources. *ACS Nano* **5**, 3385-3390 (2011).

R8. Sun, Z. et al. Growth of graphene from solid carbon sources. *Nature* **468**, 549-552 (2010).

R9. Ji, H. et al. Graphene growth using a solid carbon feedstock and hydrogen. *ACS Nano* **5**, 7656-7661 (2011).

R10. Kwak, J. et al. In situ observations of gas phase dynamics during graphene growth using solid-state carbon sources. *Phys. Chem. Chem. Phys.* **15**, 10446 (2013).

R11. Li, X., Kurasch, S., Kaiser, U. & Antonietti, M. Synthesis of Monolayer-Patched Graphene from Glucose. *Angew. Chem. Int. Edit.* **51**, 9689-9692 (2012).

R12. Ferrari, A.C. Raman spectroscopy of graphene and graphite: Disorder, electron-phonon coupling, doping and nonadiabatic effects. *Solid State Commun.* **143**, 47-57 (2007).

R13. Zhang, X. et al. Nitrogen-doped graphene network supported copper nanoparticles encapsulated with graphene shells for surface-enhanced Raman scattering. *Nanoscale* **7**, 17079-17087 (2015).

R14. Zhang, B., Song, J., Yang, G. & Han, B. Large-scale production of high-quality graphene using glucose and ferric chloride. *Chem. Sci* **5**, 4656-466 (2014).

R15. Wang, X. et al. Three-dimensional strutted graphene grown by substrate-free sugar blowing for high-power-density supercapacitors. *Nat. Commun.* **4**, 2905 (2013).

R16. Zhao, W. et al. Defect-engineered heat transport in graphene: A route to high efficient thermal rectification. *Sci. Rep.* **5**, 11962 (2015).

R17. Balandin, A.A. et al. Superior thermal conductivity of single-layer graphene.

Nano Lett. **8**, 902-907 (2008).

R18. Yoon, J., Lee, J., Kim, S., Kim, K. & Jang, J. Three-dimensional graphene nano-networks with high quality and mass production capability via precursor-assisted chemical vapor deposition. *Sci. Rep.* **3**, 1788 (2013).

R19. Cançado, L.G. et al. Quantifying defects in graphene via Raman spectroscopy at different excitation energies. *Nano Lett.* **11**, 3190-3196 (2011).

R20. Malekpour, H. et al. Thermal conductivity of graphene with defects induced by electron beam irradiation. *Nanoscale* **8**, 14608-14616 (2016).

R21. Hsieh, C. & Liu, W. Synthesis and characterization of nitrogen-doped graphene nanosheets/copper composite film for thermal dissipation. *Carbon* **118**, 1-7 (2017).

R22. Zheng, L. et al. N-doped graphene-based copper nanocomposite with ultralow electrical resistivity and high thermal conductivity. *Sci. Rep.* **8**, 9248 (2018).

R23. Luo, B. et al. Fabrication of lignin-based nano carbon film-copper foil composite with enhanced thermal conductivity. *Nanomater.* **9**, 1681 (2019).

R24. Wejrzanowski, T. et al. Thermal conductivity of metal-graphene composites. *Mater. Design* **99**, 163-173 (2016).

R25. Chu, K. et al. Largely enhanced thermal conductivity of graphene/copper composites with highly aligned graphene network. *Carbon* **127**, 102-112 (2018).

R26. Mehta, R., Chugh, S. & Chen, Z. Enhanced electrical and thermal conduction in graphene-encapsulated copper nanowires. *Nano Lett.* **15**, 2024-2030 (2015).

R27. Li, J. et al. Highly thermally conductive graphene film produced using glucose under low-temperature thermal annealing. *J. Mater. Sci.* **54**, 7553-7562 (2019).

R28. Yang, W. et al. Carbon nitride template-directed fabrication of nitrogen-rich porous graphene-like carbon for high performance supercapacitors. *Carbon* **130**, 325-332 (2018).

R29. Lee, J., Kim, S., Yoon, J. & Jang, J. Chemical vapor deposition of mesoporous graphene nanoballs for supercapacitor. *ACS Nano* **7**, 6047-6055 (2013).

R30. Goli, P. et al. Thermal properties of graphene-copper-graphene

heterogeneous films. *Nano Lett.* **14**, 1497-1503 (2014).

R31. Kim, Y.D., Oh, N.L., Oh, S. & Moon, I. Thermal conductivity of W–Cu composites at various temperatures. *Mater. Lett.* **51**, 420-424 (2001).

R32. Shinoda, W., Shiga, M. & Mikami, M. Rapid estimation of elastic constants by molecular dynamics simulation under constant stress. *Phys. Rev. B* **69**, 134103 (2004).

R33. Martyna, G.J., Tobias, D.J. & Klein, M.L. Constant pressure molecular dynamics algorithms. *J. Chem. Phys.* **101**, 4177-4189 (1994).

R34. Parrinello, M. & Rahman, A. Polymorphic transitions in single crystals: A new molecular dynamics method. *J. Appl. Phys.* **52**, 7182-7190 (1981).

REVIEWERS' COMMENTS:

Reviewer #1 (Remarks to the Author):

I am satisfied that the authors have addressed my comments, and consider the article appropriate for publication in Nature Communications.

Reviewer #2 (Remarks to the Author):

The authors improved the manuscript. Publication is recommended.

Responses to the Reviewers' Comments

Reviewer #1 (Remarks to the Author):

I am satisfied that the authors have addressed my comments, and consider the article appropriate for publication in Nature Communications.

Response: We thank the reviewer very much for the positive comments.

Reviewer #2 (Remarks to the Author):

The authors improved the manuscript. Publication is recommended.

Response: We thank the reviewer very much for the positive comments.